# Multilevel Generative Samplers for Investigating Critical Phenomena

**Ankur Singha**[1,2*], **Elia Cellini**[3,4*], **Kim A. Nicoli**[5,6*]
**Karl Jansen**[7], **Stefan Kühn**[7], **Shinichi Nakajima**[1,2,8]

[1]BIFOLD, Germany, [2] Technische Universität Berlin, Germany
[3]Università degli Studi di Torino, Italy, [4] INFN Torino, Italy, [5]University of Bonn, Germany
[6]Helmholtz Institute for Radiation and Nuclear Physics (HISKP)
[7]Deutsches Elektronen-Synchrotron (DESY), Germany, [8]RIKEN Center for AIP, Japan

## Abstract

Investigating critical phenomena or phase transitions is of high interest in physics and chemistry, for which Monte Carlo (MC) simulations, a crucial tool for numerically analyzing macroscopic properties of given systems, are often hindered by an emerging divergence of correlation length—known as *scale invariance at criticality* (SIC) in the renormalization group theory. SIC causes the system to behave the same at any length scale, from which many existing sampling methods suffer: long-range correlations cause critical slowing down in Markov chain Monte Carlo (MCMC), and require intractably large receptive fields for generative samplers. In this paper, we propose a Renormalization-informed Generative Critical Sampler (RiGCS)—a novel sampler specialized for near-critical systems, where SIC is leveraged as an advantage rather than a nuisance. Specifically, RiGCS builds on MultiLevel Monte Carlo (MLMC) with Heat Bath (HB) algorithms, which perform ancestral sampling from low-resolution to high-resolution lattice configurations with site-wise-independent conditional HB sampling. Although MLMC-HB is highly efficient under exact SIC, it suffers from a low acceptance rate under slight SIC violation. Notably, SIC violation always occurs in finite-size systems, and may induce long-range and higher-order interactions in the renormalized distributions, which are not considered by independent HB samplers. RiGCS enhances MLMC-HB by replacing a part of the conditional HB sampler with generative models that capture those residual interactions and improve the sampling efficiency. Our experiments show that the effective sample size of RiGCS is a few orders of magnitude higher than state-of-the-art generative model baselines in sampling configurations for $128 \times 128$ two-dimensional Ising systems. SIC also allows us to adopt a specialized sequential training protocol with model transfer, which significantly accelerates training.

## 1 Introduction

Monte Carlo (MC) simulations, where samples from a Boltzmann distribution are used to estimate macroscopic properties, are ubiquitous in many fields of science, ranging from chemistry (Metropolis et al., 1953), statistical physics (Hastings, 1970; Creutz et al., 1983), and quantum field theory (Creutz et al., 1979; Wilson, 1980) to biology (Huelsenbeck et al., 2001) and financial analysis (Doucet et al., 2001). In MC simulations, the ability to efficiently sample from unnormalized distributions in a high-dimensional space poses crucial challenges. Standard algorithms such as Monte Carlo Markov Chain (MCMC) methods (Metropolis & Ulam, 1949; Robert & Casella, 2004) are often plagued by, e.g., slow convergence (Cowles & Carlin, 1996), energy barriers and local minima (Cérou et al., 2012), and critical slowing down (Wolff, 1990; 2004; Schaefer et al., 2011). This paper specifically tackles the problem of critical slowing down around critical regimes. In the broader context of physical sciences, the term *criticality* refers to situations where a system

---

*Equal contributions. Correspondence to {`a.singha,nakajima`}@tu-berlin.de

undergoes a sharp behavioral change, often associated with phase transitions (Nishimori & Ortiz, 2011). At criticality, physical systems typically exhibit self-similarity with respect to the change of scale, i.e., the physics of the coarse-grained system is similar to that of the fine-grained one. This phenomenon, called *scale invariance at criticality* (SIC), requires us to deal with arbitrarily long-range correlations, for which standard MCMC samplers with local moves undergo critical slowing down with arbitrarily long integrated auto-correlation time. Although many highly specialized cluster algorithms, leveraging non-local moves, have been developed in the context of spin systems (Wolff, 1989b;a), critical slowing down still represents one of the major shortcomings of MCMC approaches.

For efficient sampling around criticality, *MultiLevel* (or Multiscale) Monte Carlo with Heat Bath (MLMC-HB) algorithms (Schmidt, 1983; Faas & Hilhorst, 1986; Jansen et al., 2020) were developed, based on *Renormalization Group Theory*(RGT) (Kadanoff, 1966; Wilson, 1971; Wilson & Kogut, 1974; Cardy, 1996). RGT systematically analyzes how macroscopic features emerge when the system is *coarse-grained* to larger length scales by marginalizing the degrees of freedom corresponding to the fine lattice grid, and provides crucial insights into critical phenomena of physical systems (Wilson, 1971; Fisher, 1973; Shankar, 1994; Cardy, 1996). An important outcome of RGT is the emergence of SIC over the coarse-grained and fine-grained lattices. Adopting the block-spin transformations (Kadanoff, 1966) for partitioning of lattice sites, Schmidt (1983) proposed MLMC-HB that performs ancestral sampling from the coarsest lattice sites to the finer ones. Crucially, conditional distributions between consecutive resolution levels can thus be factorized into independent distributions under SIC, for which sampling can be efficiently performed by HB algorithms. In MLMC-HB, the long-range correlations are captured in the low resolution lattice, which is much easier than capturing them in the original high resolution lattice.

Machine learning techniques are also seen as potential candidates to, either partially or fully, overcome the shortcomings of MCMC algorithms. In particular, generative models with accessibility to exact sampling probabilities—such as normalizing flows (Rezende & Mohamed, 2015; Kobyzev et al., 2020; Papamakarios et al., 2021) and autoregressive models (van den Oord et al., 2016c;b; Salimans et al., 2017)—offer efficient independent sampling and unbiased MC estimation via importance sampling, showing notable success across various domains. Such applications include statistical physics (Wu et al., 2019; Nicoli et al., 2020), quantum many-body systems (Hibat-Allah et al., 2020), quantum chemistry (Noé et al., 2019; Gebauer et al., 2019), string theory (Caselle et al., 2024), and lattice field theory (Albergo et al., 2019; Nicoli et al., 2021; Caselle et al., 2022; Cranmer et al., 2023; Abbott et al., 2024). However, capturing long-range interactions in large lattice systems may require intractably large receptive fields. Generative models therefore tend to struggle to generate samples around criticality, except for a few recent works whose goal was to mitigate critical slowing down (Pawlowski & Urban, 2020; Białas et al., 2023).

In this work, we propose a Renormalization-informed Generative Critical Sampler (RiGCS), which enhances MLMC-HB algorithms by mitigating their major weakness—i.e., the HB samplers in MLMC-HB ignore long-range and higher-order interactions that may exist in renormalized systems when SIC does not hold exactly. Instead, RiGCS uses generative models with sufficiently large receptive fields to approximate renormalized distributions, effectively capturing the majority of those residual interactions. In our experiments, RiGCS drastically improves the sampling efficiency of MLMC-HB, and achieves an effective sample size a few orders of magnitude higher than the previous state-of-the-art generative sampler (Białas et al., 2023) for the two-dimensional Ising model. Furthermore, we propose a specialized sequential training procedure with *warm starts*, which transfers model parameters between different resolution levels, substantially improving the training efficiency. Our contributions include:

- **Renormalization-informed Multilevel Sampling**: We propose a novel method that leverages both SIC and generative modeling to efficiently draw samples from Boltzmann distributions around criticality.

- **Sequential Training with Warm Starts**: We propose a sequential training procedure that initially samples a small-scale system and progressively transitions to the target large-scale system. Model parameters trained on smaller systems are transferred to larger ones for initialization, enabling a warm start that significantly accelerates training.

As with most generative neural samplers for simulating lattice-based physical systems, we do not claim that our method surpasses state-of-the-art MCMC samplers, such as cluster methods for the Ising model, which remain unmatched for general observable estimation in large-scale systems.

**Related Work** Renormalization Group Theory (RGT) (Wilson, 1971; Wilson & Kogut, 1974; Kadanoff, 1966) has profoundly influenced the study of critical behavior in statistical systems and quantum field theory. Leveraging results of RGT, MLMC-HB for near-critical systems was proposed (Schmidt, 1983), showing notable improvements in sampling efficiency for one- and two-dimensional Ising models. Adopting a particular partitioning of lattice sites, called *block-spin transformations* (Kadanoff, 1966), MLMC-HB draws samples hierarchically by the site-wise independent conditional HB sampling based on the renormalized systems at different scales. Faas & Hilhorst (1986) further enhanced MLMC-HB by incorporating long-range interactions. Recently, Jansen et al. (2020) introduced a low variance MC estimator by leveraging the correlations between the lattices with consecutive resolution levels, which further advanced MLMC-HB.

A variety of generative models, including Generative Adversarial Networks (GANs) (Pawlowski & Urban, 2020; Singha et al., 2022), Variational Auto Encoders (VAEs) (D'Angelo & Böttcher, 2020), and energy-based models (D'Angelo & Böttcher, 2020; Torlai & Melko, 2016), have been used as independent MC samplers for lattice systems. Generative models with accessibility to the exact sampling probability are particularly useful for MC simulations, because they allow for unbiased MC estimation by importance sampling or neural MC (Nicoli et al., 2020), i.e., sampling with the Metropolis-Hastings rejection. Specifically, Variational Autoregressive Networks (VANs) (Wu et al., 2019; Nicoli et al., 2020) and normalizing flows (Albergo et al., 2019) have proven to be highly effective for discrete and continuous systems, respectively. Recent works (Singha et al., 2023a;b; Gerdes et al., 2023) introduced conditional normalizing flows for scalar field and gauge theories, showing that models trained away from criticality can interpolate (or extrapolate) for drawing samples near criticality. Nicoli et al. (2021) and Bulgarelli et al. (2024) demonstrated that generative models are particularly useful for estimating thermodynamic observables, e.g., free energy and entropy, which cannot be directly estimated with standard MCMC methods.

Recently, hierarchical sampling approaches have been integrated with generative modeling both for discrete systems (Li & Wang, 2018; Białas et al., 2022) and continuous lattice field theories (Finkenrath, 2024; Abbott et al., 2024). Neural Network Renormalization Group (NeuralRG) (Li & Wang, 2018) uses a hierarchical bijective map to learn a renormalization transform, and was applied to the Ising model using a continuous relaxation technique. Hierarchical Autoregressive Network (HAN) (Białas et al., 2022)—a state-of-the-art generative sampler for discrete physical systems— uses a recursive domain decomposition (Cè et al., 2016), and performs independent conditional sampling for separate regions with trained VANs. The HAN approach has shown improved sampling efficiency compared to MLMC-HB in two-dimensional Ising models. We refer to Appendix A for an extended review of related works.

## 2 BACKGROUND

This paper focuses on Monte Carlo (MC) simulations of hypercubic lattice systems around criticality. We refer to a (row vector) *sample* $s \in \mathcal{S}^V$ to be a *configuration* on the $V = N^D$ grid points in the $D$ dimensional lattice, where $\mathcal{S}$ denotes the domain of the random variable at each site, and $N$ denotes the lattice size (per dimension). Given a Hamiltonian (or energy) $H(s)$ describing the interactions between the lattice sites, MC simulations draw samples from the Boltzmann distribution

$$p(s) = \tfrac{1}{Z} e^{-\beta H(s)}, \tag{1}$$

where $\beta$ is a constant inversely proportional to the temperature, and $Z$ is the (typically unknown) normalization constant called the partition function. With a sufficient number $M$ of samples, physical observables $\mathcal{O}$, e.g., energy, magnetization, and susceptibility, can be estimated by averaging over the sample configurations $\langle \mathcal{O} \rangle \approx \frac{1}{M} \sum_{m=1}^{M} \mathcal{O}(s_m)$, thus revealing macroscopic physical properties and phenomena like phase transitions. Below, we introduce common sampling methods with and without machine learning techniques.

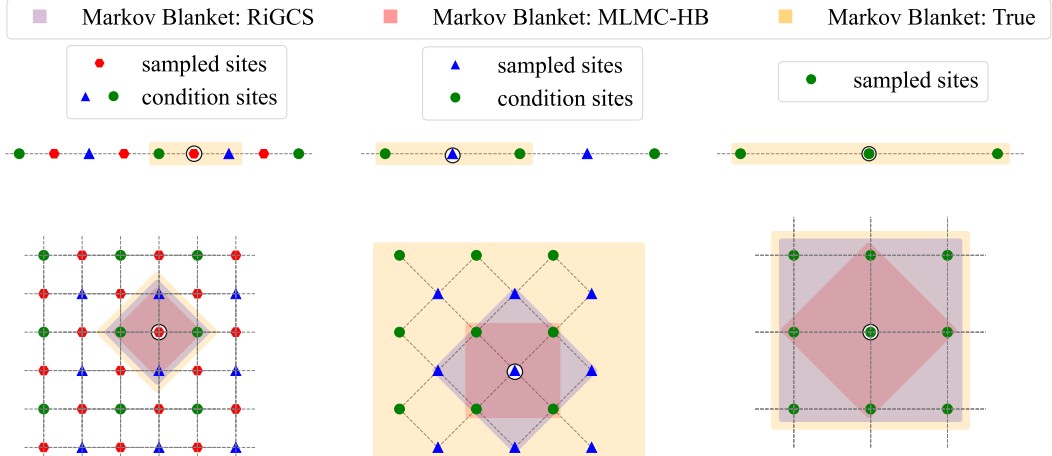

Figure 1: Site partitionings based on the block-spin transformation (Kadanoff, 1966). The ancestral sampling is performed from the coarsest level (right) to the finest level (left), namely, in the order of $s^{L-2}$ (green), $s^{L-1}$ (blue), and $s^L$ (red). In the one-dimensional case (upper row), the marginal distribution at each resolution level involves only nearest neighbor (NN) interactions. Consequently, the true Markov blanket (indicated by yellow shadows) of the highlighted site (black circle) includes only NN conditioning sites, enabling precise and independent HB sampling. In the two-dimensional case (lower row), marginal distributions generally include long-range and higher-order interactions. As a result, except at the finest level, the true Markov blanket (yellow shadows) possibly encompasses all other sites. While MLMC-HB with NN Markov blankets (red shadows) can still serve as an approximate trial sampler, its locality leads to low acceptance rates for large lattice sizes. In contrast, our RiGCS, with wider Markov blankets (purple shadows) induced by the receptive field of generative models, effectively captures long-range and higher-order interactions, providing more accurate samples. Note that the Markov blankets of RiGCS used for the intermediate resolutions in our implementation are of the size $7 \times 7$ (i.e., much larger than the one shown in this figure). Sites at $s_{L-2}$ (green) can be further partitioned until the dimension of $s^0$ gets sufficiently small.

## 2.1 MARKOV CHAIN MONTE CARLO (MCMC) METHODS AROUND CRITICALITY

Markov Chain Monte Carlo (MCMC) methods are a class of algorithms used to sample from unnormalized distributions. Since the partition function is not analytically computable in most physical systems, MCMC methods are fundamental tools for performing MC simulations in, e.g., statistical mechanics and lattice quantum field theory. A crucial challenge for MCMC sampling around criticality is to cope with long-range correlations. When distant regions in the lattice become strongly correlated, the standard MCMC methods that rely on local updates struggle to move from a low-energy state to another low-energy state. This is because local updates generally ignore the correlations, and thus, the proposed trials tend to be rejected in the Metropolis-Hastings rejection step due to the increased energy. This results in a long autocorrelation time for the Markov chain—a phenomenon known as *critical slowing down* (Wolff, 1990). Two approaches were developed to mitigate critical slowing down.

**Cluster algorithms** Cluster algorithms (Wolff, 1989a; Swendsen & Wang, 1987) perform global updates by identifying clusters of correlated lattice sites and flipping them collectively. These global updates efficiently reduce the autocorrelation time and mitigate critical slowing down. Appendix B introduces two variants of cluster methods, including the Wolff algorithm (Wolff, 1989a) which we refer to as the *cluster algorithm* throughout the rest of the paper. Cluster methods are not seen as universal remedies against critical slowing down because they are not generally applicable to arbitrary continuous variable systems, although generalizations to specific continuous systems were proposed (Kent-Dobias & Sethna, 2018).

**Multilevel Monte Carlo with Heat Bath (MLMC-HB)** The MultiLevel Monte Carlo with Heat Bath (MLMC-HB) algorithm (Schmidt, 1983)—a protocol inspired by RGT—was developed for sampling physical systems around criticality. We denote by $\boldsymbol{J}^{\mathrm{NN}} \in \mathbb{R}^{V \times V}$ a homogeneous $2 \cdot D$ nearest neighbor (NN) interaction matrix that satisfies

$$(\boldsymbol{J}^{\mathrm{NN}})_{v,v'} = \begin{cases} J & \text{if } (v, v') \text{ are } 2 \cdot D \text{ nearest neighbor pairs,} \\ 0 & \text{otherwise,} \end{cases} \tag{2}$$

for $J \in \mathbb{R}$, where $(\cdot)_{v,\cdots}$ denotes the $(v, \cdots)$-th entry of a vector, matrix, or tensor. For some important physical systems, including the Ising model (Onsager, 1944) and the XY model (Kosterlitz, 1974), the Hamiltonian can be written as

$$H(\boldsymbol{s}) = -\boldsymbol{s}\boldsymbol{J}^{\mathrm{NN}}\boldsymbol{s}^{\top} \tag{3}$$

with $\boldsymbol{J}^{\mathrm{NN}} \in \mathbb{R}^{V \times V}$.[1] In MLMC-HB, one partitions the lattice sites into $L + 1$ levels $\boldsymbol{s} = (\boldsymbol{s}^L, \boldsymbol{s}^{L-1}, \ldots, \boldsymbol{s}^0)$ so that all $2 \cdot D$ nearest neighbors of each entry of $\boldsymbol{s}^l$ (for $l = 1, \ldots, L$) belong to the coarser level partitions $(\boldsymbol{s}^{l-1}, \ldots, \boldsymbol{s}^0)$. Figure 1 shows examples of site partitionings based on the block-spin transformations (Kadanoff, 1966) for $(D = 1)$- and $(D = 2)$-dimensional lattices, where the sites with the same color (red, blue, or green) belong to the same partition $(\boldsymbol{s}^L, \boldsymbol{s}^{L-1}, \text{ or } \boldsymbol{s}^{L-2})$. For compact descriptions, we use inequalities to express subsets of the partitions, e.g., $\boldsymbol{s}^{\leq l} = (\boldsymbol{s}^l, \ldots, \boldsymbol{s}^0)$, and $\boldsymbol{s}^{> l} = (\boldsymbol{s}^L, \ldots, \boldsymbol{s}^{l+1})$. We denote by $V_l$ the dimension of $\boldsymbol{s}^{\leq l}$, i.e., $\boldsymbol{s}^{\leq l} \in \mathbb{R}^{V_l}$ and $\boldsymbol{s}^{> l} \in \mathbb{R}^{V - V_l}$. The marginal distribution of the $l$-th level lattice is given by

$$p(\boldsymbol{s}^{\leq l}) = \int p(\boldsymbol{s})\mathcal{D}[\boldsymbol{s}^{> l}] \equiv \tfrac{1}{Z_l}e^{-\beta H_l(\boldsymbol{s}^{\leq l})}, \tag{4}$$

where the corresponding Hamiltonian $H_l(\boldsymbol{s}^{\leq l})$ (i.e., a scaled negative log-marginal probability) is called a *renormalized Hamiltonian*. An important result in RGT is that, around the criticality, the renormalized Hamiltonian for $l = 0, \ldots, \widetilde{L}$, where $\widetilde{L}$ is a few levels smaller than $L$, can be approximated as a Hamiltonian with NN interactions, i.e.,

$$H_l(\boldsymbol{s}^{\leq l}) \approx \widetilde{H}_l(\boldsymbol{s}^{\leq l}), \qquad \text{where} \qquad \widetilde{H}_l(\boldsymbol{s}^{\leq l}) = -\boldsymbol{s}^{\leq l}\boldsymbol{J}_l^{\mathrm{NN}}(\boldsymbol{s}^{\leq l})^{\top} \tag{5}$$

with the NN interaction matrix $\boldsymbol{J}_l^{\mathrm{NN}} \in \mathbb{R}^{V_l \times V_l}$.[2] If Eq. (5) holds exactly, the conditional probability $p(\boldsymbol{s}^l | \boldsymbol{s}^{\leq l-1})$ can be decomposed as

$$p(\boldsymbol{s}^l | \boldsymbol{s}^{\leq l-1}) = \prod_{v=1}^{V_l} p(s_v^l | \boldsymbol{s}^{\leq l-1}), \tag{6}$$

because the Markov blanket[3] (Bishop, 2006) of $s_v^l$ does not contain $s_{v'}^l$ for any $v' \neq v$ (see yellow shadows in the one-dimensional example in Figure 1, upper row). This makes the sampling from the conditional distribution (6) extremely easy and efficient—one can apply the HB conditional sampling *exactly* for the discrete domain (with a probability table of size $|\mathcal{S}|$), and *approximately* for the continuous domain (with, e.g., a one-dimensional Gaussian mixture). Therefore, starting from drawing samples from $p(\boldsymbol{s}_0)$ (which can be efficiently performed by HB or MCMC if $L$ is sufficiently large and hence $V_0$ is small), the ancestral sampling of the full lattice according to

$$p(\boldsymbol{s}) = \left(\prod_{l=1}^{L} p(\boldsymbol{s}^l | \boldsymbol{s}^{\leq l-1})\right) p(\boldsymbol{s}^0) \tag{7}$$

can be efficiently performed. Intuitively, MLMC-HB captures the long-range correlations by the coarse level marginals, i.e., $p(\boldsymbol{s}^{\leq l})$ for small $l$, which avoids two major difficulties—large lattice size and long-range correlations—arising at the same time.

---

[1] The whole discussion in this paper can be applied to a slight generalization with the Hamiltonian in the form of $H(\boldsymbol{s}) = -\sum_{v,v'}(\boldsymbol{J}^{\mathrm{NN}})_{v,v'}\psi(s_v, s_{v'})$, where $\psi(s, s')$ is a similarity function between two states, e.g., $\psi(s, s') = \delta_{s,s'}$ for Potts model (Wu, 1982).

[2] The corresponding NN interaction coefficient—$J$ in Eq. (2) which we refer to as $J_l$—can be analytically computed in RGT (see Appendix C). When $N, L \to \infty$, $J_l$ converges to a limiting value as $l$ decreases, and thus the *scale invariance at criticality* (SIC) emerges. Since we consider finite lattices, exact SIC never holds. Nevertheless, in one-dimensional finite lattices, the renormalized Hamiltonians consist only of NN interactions, which is sufficient for MLMC-HB to be accurate, as explained in Figure 1.

[3] The Markov blanket of a random variable $s_v$ is a set of other random variables $\mathcal{B}_{s_v} \subseteq \{s_{v'}\}_{v' \neq v}$ that have sufficient information to determine the conditional distribution of $s_v$ given the other random variables, i.e., $p(s_v | \mathcal{B}_{s_v}) = p(s_v | \{s_{v'}\}_{v' \neq v})$.

For the $(D = 1)$-dimensional lattice (see Figure 1 upper row), it is known that Eq. (5), and thus Eq. (6), hold exactly, and therefore, MLMC-HB generates accurate samples from the target Boltzmann distribution. For $D \geq 2$ (see Figure 1 lower row), Eq. (5) holds only approximately, and therefore, MLMC-HB should be combined with importance sampling or Metropolis-Hastings rejection. Namely, one should draw samples according to

$$q^{\text{NN}}(\boldsymbol{s}) = p(\boldsymbol{s}^L | \boldsymbol{s}^{\leq L-1}) \left( \prod_{l=1}^{L-1} q^{\text{NN}}(\boldsymbol{s}^l | \boldsymbol{s}^{\leq l-1}) \right) q^{\text{NN}}(\boldsymbol{s}^0), \tag{8}$$

and compensate the sampling bias by using the sampling probability $q^{\text{NN}}(\boldsymbol{s})$, where $\{q^{\text{NN}}(\boldsymbol{s}^l | \boldsymbol{s}^{\leq l-1})\}_{l=1}^{L-1}$ and $q^{\text{NN}}(\boldsymbol{s}_0)$ are approximate distributions with NN interactions to the true conditionals $\{p(\boldsymbol{s}^l | \boldsymbol{s}^{\leq l-1})\}_{l=1}^{L-1}$ and the true marginal $p(\boldsymbol{s}_0)$, respectively. Unfortunately, the approximation errors accumulate through ancestral sampling, leading to a significantly low acceptance rate for large lattice sizes. We will show in Section 4 that MLMC-HB is not very efficient for $D = 2$. Further details on RGT and MLMC-HB are given in Appendix C and Appendix D, respectively.

## 2.2 Generative Modeling for MC Simulations

In recent years, deep generative models have gained significant traction in the field of physics for their efficient modeling of complicated probability distributions. In particular, normalizing flows (Kobyzev et al., 2020) and autoregressive neural networks (van den Oord et al., 2016c) became very popular in the context of computational physics due to their intrinsic capability of providing the exact sampling probability $q_{\boldsymbol{\theta}}(\boldsymbol{s})$, which allows *asymptotically* unbiased MC estimation. Notably, well-trained generative models can provide *independent* samples from an approximate distribution, and asymptotically unbiased estimates of physical observables can then be computed by importance sampling (Nicoli et al., 2020): $\langle \mathcal{O} \rangle \approx \frac{1}{M} \sum_{m=1}^{M} \frac{\widetilde{w}_m}{\sum_{m'=1}^{M} \widetilde{w}_{m'}} \mathcal{O}(\boldsymbol{s}_m)$, where $\widetilde{w}_m = e^{-\beta H(\boldsymbol{s}_m)}/q_{\boldsymbol{\theta}}(\boldsymbol{s}_m)$ are the unnormalized importance weights. However, naive generative modeling can be problematic for sampling large lattices near criticality because large receptive fields are required to capture long-range correlations. Improving the scalability of generative samplers for large systems is therefore one of the most crucial challenges to achieving the same level of performance as state-of-the-art MCMC samplers, and ultimately surpassing them in efficiency.

## 3 Method

In this section, we describe our proposed method that enhances MLMC-HB (introduced in Section 2.1) with generative modeling (introduced in Section 2.2). We focus on $(D = 2)$-dimensional lattice systems with $(V = N \times N)$ grid points.

### 3.1 Renormalization-informed Generative Critical Sampler (RiGCS)

Higher-order RGT (Maris & Kadanoff, 1978) shows that, for $D = 2$, the Hamiltonian of the marginal distribution (4) consists not only of the NN interaction terms but also of long-range and higher-order interaction terms:

$$H_l(\boldsymbol{s}^{\leq l}) = -\boldsymbol{s}^{\leq l} \boldsymbol{J}_l^{\text{NN}}(\boldsymbol{s}^{\leq l})^\top - \boldsymbol{s}^{\leq l} \boldsymbol{J}_l^{\text{LR}}(\boldsymbol{s}^{\leq l})^\top - \sum_{v,v',v''} (\mathcal{J}_l^{\text{HO}})_{v,v',v''} s_v^{\leq l} s_{v'}^{\leq l} s_{v''}^{\leq l} + \cdots, \tag{9}$$

where $\boldsymbol{J}_l^{\text{LR}}$ and $\mathcal{J}_l^{\text{HO}}$ denote the matrix and the tensor that express the long-range and high-order interactions, respectively.[4] Instead of simply approximating the renormalized Hamiltonian (9) by the Hamiltonian (5) with NN interactions (as done in MLMC-HB), our method, called Renormalization-informed Generative Critical Sampler (RiGCS), approximates it with generative models that can capture the long-range and higher-order interactions. Specifically, RiGCS performs ancestral sampling according to

$$q_{\boldsymbol{\theta}}(\boldsymbol{s}) = p(\boldsymbol{s}^L | \boldsymbol{s}^{\leq L-1}) \left( \prod_{l=1}^{L-1} q_{\boldsymbol{\theta}_l}(\boldsymbol{s}^l | \boldsymbol{s}^{\leq l-1}) \right) q_{\boldsymbol{\theta}_0}(\boldsymbol{s}^0), \tag{10}$$

---

[4]Note the difference between the "interactions" and the "correlations". The former means the direct cross dependent terms in the Hamiltonian, while the latter means statistical dependence in the Boltzmann distribution.

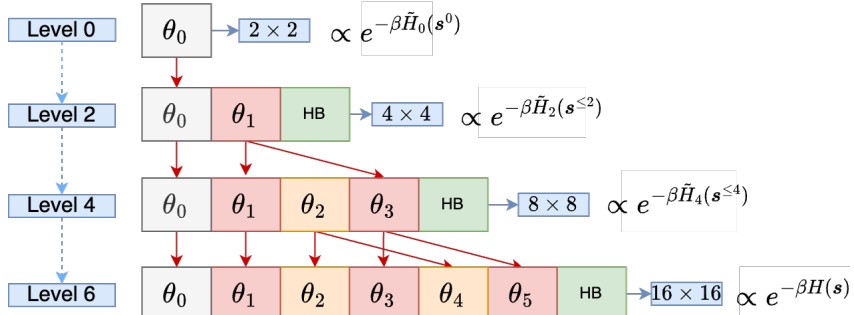

Figure 2: Illustration of sequential training for a six-layer ($L = 6$) RiGCS model on a $16 \times 16$ lattice. Training begins with the marginal model $q_{\theta_0}(s^0)$ on the smallest $2 \times 2$ target Boltzmann distribution. Conditional models for progressively larger RiGCS layers are then trained sequentially on target Boltzmann distributions for increasingly larger lattices. Red arrows indicate model transfer initializations, while parameters without incoming red arrows are randomly initialized. For larger lattices, the final step is repeated until the target size is achieved.

where $q_{\theta_l}(s^l|s^{\leq l-1})$ for $l = L-1, \ldots, 1$ and $q_{\theta_0}(s_0)$ are conditional and unconditional generative models that approximate $p(s^l|s^{\leq l-1})$ and $p(s_0)$, respectively. Here, $\theta = (\theta_{L-1}, \ldots, \theta_0)$ denotes all model's trainable parameters. Similarly to the configuration variable $s$, we use inequalities to express subsets of the parameters, e.g., $\theta_{\leq l} = (\theta_l, \ldots, \theta_0)$. Note that, at the finest level, the conditional distribution $p(s^L|s^{\leq L-1})$ has only NN interactions by assumption, and therefore, the exact HB algorithm can be efficiently applied without generative modeling. Therefore, it holds that $\theta = \theta_{\leq L} = \theta_{\leq L-1}$—as there is no model parameter at the $L$-th level.

## 3.2 RECEPTIVE FIELD DESIGN

It is known that long-range and higher-order interactions between lattice sites diminish with increasing distance (Maris & Kadanoff, 1978). Therefore, the accuracy of approximating the renormalized Hamiltonian can be controlled by adjusting the receptive field size of conditional generative models—the larger the receptive field is, the more accurate the approximation to the renormalized Hamiltonian (9). Compared to vanilla generative models (without multilevel sampling), where the receptive field needs to cover the whole correlation range in the original finest-level lattice, our RiGCS approach allows us to keep the receptive field of each conditional model small. In particular, if we set the number of levels $L$ proportional to the lattice size $N$ (per dimension), we can keep the receptive field size constant for different $N$. This is because RiGCS captures the long-range interactions in the coarser levels—the receptive field size $\alpha \times \alpha$ of the coarsest generative model $q_{\theta_0}(s^0)$ effectively amounts to the receptive field size $\alpha L/2 \times \alpha L/2$ in the finest lattice. Although the optimal receptive field size for each level $l$ should in principle exist such that the accumulated approximation error is minimized for a given computational cost, we use the same architecture for the conditional models for $l = L-1, \ldots, 1$ in this work. This choice, with makes the model complexity of RiGCS linear to $L$, enables efficient model transfer in training, as explained below.

## 3.3 SEQUENTIAL TRAINING WITH MODEL TRANSFER INITIALIZATION

We train our RiGCS by minimizing the reverse Kullback-Leibler (KL) divergence, i.e.,

$$\min_{\theta} \mathrm{KL}(q_\theta(s)\|p(s)) = \sum_{s \in \mathcal{S}^V} q_\theta(s) \log \frac{q_\theta(s)}{p(s)} \approx \frac{1}{M_{\mathrm{tr}}} \sum_{m=1}^{M_{\mathrm{tr}}} \log \frac{q_\theta(s_m)}{p(s_m)}, \quad (11)$$

which is estimated with the generated samples $\{s_m \sim q_\theta(s)\}_{m=1}^{M_{\mathrm{tr}}}$—training data drawn from the target distribution are not required. However, training all parameters $\theta$ from scratch, e.g., with random initialization, tends to suffer from long initial random walking steps. This is because the randomly initialized RiGCS generates random samples, with which the estimated stochastic gradient of the objective (11) rarely provides useful signal to train the model. We tackle this problem with a specific training procedure with *model transfer*, again based on RGT. We choose $L$ to be an even number, and consider a set of *sequential target* Boltzmann distributions $p_{L'}(s^{\leq L'}) \propto e^{-\beta \widetilde{H}_{L'}(s^{\leq L'})}$

for $L' = 0, 2, 4, \ldots, L$, where $\{\widetilde{H}_l(s^{\leq l})\}_{l=0}^L$ are the approximate renormalized Hamiltonians with NN interactions, defined in Eq.(5), and $\widetilde{H}_L(s^{\leq L}) = H(s)$. For each target $p_{L'}(s^{\leq L'})$ in the increasing order of $L'$, we train a RiGCS $q_{\theta^{\leq L'-1}}(s^{\leq L'})$ that shares the same coarsest lattice size $V_0$ as the model, $q_{\theta^{\leq L-1}}(s^{\leq L}) = q_\theta(s)$, for the original target distribution. This allows for initializing the RiGCS parameters for learning $p_{L'}(s^{\leq L'})$ with the corresponding parameters already trained on $p_{L'-2}(s^{\leq L'-2})$—an easier (smaller lattice size) system. Figure 2 illustrates this procedure, where the initializations are indicated by the red arrows. Thanks to SIC, the models connected by the red arrows are similar to each other, as detailed in Appendix E.1. Note that all parameters except $\theta_0, \theta_1, \theta_2$—which are trained on the three smallest lattice sizes with random initializations—can be *warm-started*. The pseudocode for the sampling and training routines of RiGCS is provided in Appendix E.2.

## 4 NUMERICAL EXPERIMENTS

We evaluate our proposed RiGCS and compare it against several baseline methods. The experimental setup is detailed below. The code is available at https://github.com/mlneuralsampler/multilevel.

**Target Physical Systems**   We adopt the two-dimensional Ising model, for which the Hamiltonian is given by Eq. (3) with the 2-dimensional binary lattice, i.e., $s \in \mathcal{S}^V = \{-1, 1\}^{N \times N}$, and $J = 1$. This commonly used benchmark exhibits a second-order phase transition (critical point), and is exactly solvable (Onsager, 1944)—i.e., the ground-truth is analytically computed. We set $\beta = 0.44$, which corresponds to the critical (inverse) temperature where the phase transition occurs by spontaneous symmetry breaking in the limit of an infinite lattice.

**Generative models**   Since the lattice sites are discrete random variables in a binary domain, we use autoregressive neural networks (ARNNs)

$$q_\theta(s) = \left( \prod_{v=2}^V q_\theta(s_v \mid s_{v-1}, \ldots, s_1) \right) q_\theta(s_1) \tag{12}$$

for unconditional and conditional generative models at each level of RiGCS. More specifically, we adopt PixelCNNs (van den Oord et al., 2016c;b), which allow us to control their receptive fields by adjusting the convolution kernel sizes. Further details on ARNNs are given in Appendix F.

**Baselines**   We compare our method against three different baselines, MLMC-HB (Schmidt, 1983), VAN (plain ARNN without multilevel sampling (Wu et al., 2019)), and HAN (Białas et al., 2022). We also evaluate the Wolff cluster method (see Appendix B), which is widely recognized as a highly efficient sampler for the Ising model, surpassing the performance of generative modeling approaches. In our experiments, RiGCS achieves comparable (though slightly inferior) performance to the cluster method on large lattices, while outperforming HAN—the previous state-of-the-art generative model—and other baselines.

**Model Architecture:**   The RiGCS architecture (10) consists of autoregressive models at each level. At the coarsest level ($l = 0$), the generative model $q_{\theta_0}(s^0)$ employs a PixelCNN architecture with three masked convolutional layers, each with 12 channels and a half-kernel size of 6. At all intermediate levels except the finest one, the conditional generative models $\{q_{\theta_l}(s^l|s^{\leq l-1})\}_{l=1}^{L-1}$ utilize convolutional neural networks (CNNs) with two convolutional layers, both featuring 12 channels with kernel sizes of 5 and 3, respectively. We design the kernel size to ensure that the receptive field adequately captures long-range dependencies at each scale. Consequently, the receptive field of $q_{\theta_0}(s^0)$ spans the entire coarsest lattice, ensuring comprehensive coverage. For intermediate levels ($l = 1, \ldots, L-1$), the receptive field of $q_{\theta_l}(s^l|s^{\leq l-1})$ extends over an $7 \times 7$ region, effectively capturing long-range and higher-order interactions up to sites that are three steps apart. At the finest level ($l = L$), RiGCS performs exact independent Heat-Bath (HB) sampling using $p(s^L|s^{\leq L-1})$. For further implementation details, we refer readers to Appendix G.

### 4.1 FREE ENERGY ESTIMATION

We evaluate the sampling methods in terms of the bias and the variance in estimating the free energy—a thermodynamic observable. For the cluster MCMC method, we combine it with annealed

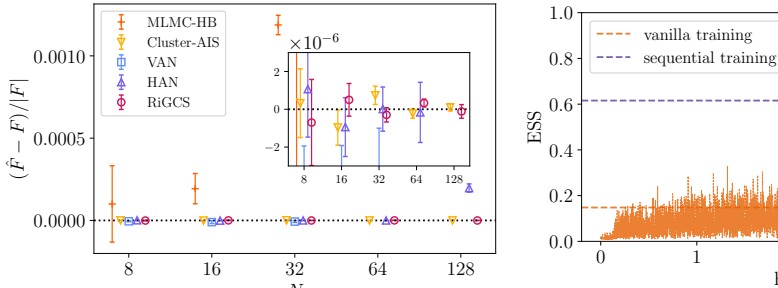

Figure 3: **Left**: Relative estimation error for the free energy. Only RiGCS provides estimates for $N = 128$ that are comparable to those of Cluster-AIS. The vanilla VAN cannot be trained for $N \geq 64$ in reasonable time. **Right**: ESS as a function of training hours for RiGCS with the vanilla training with random initialization (orange) and with the proposed sequential training (purple) for $N = 64$. The plot of the sequential training starts at $\approx 2.3$ hours when the pretraining for smaller lattice systems is finished. The ESS at each training epoch is computed with $M = 16$ samples, and the markers at "final" show the ESS computed with $M = 10^6$ at the end of training.

importance sampling (Neal, 2001) (Cluster-AIS) for estimating the free energy (Caselle et al., 2016). For generative samplers, i.e., VAN, HAN, and our RiGCS, we use the asymptotically unbiased estimator introduced by Nicoli et al. (2020):

$$\hat{F} = -\frac{1}{\beta} \log \hat{Z}, \quad \text{where} \quad \hat{Z} = \frac{1}{M} \sum_{m=1}^{M} e^{-\beta H(\boldsymbol{s}_m)} / q_{\boldsymbol{\theta}}(\boldsymbol{s}_m), \quad \boldsymbol{s}_m \sim q_{\boldsymbol{\theta}}(\boldsymbol{s}). \tag{13}$$

Here $M$ is the number of generated samples. We use this estimator also for MLMC-HB by computing all factors in Eq. (8) including the normalization constants. This is tractable because all conditionals $\{q^{\mathrm{NN}}(\boldsymbol{s}^l|\boldsymbol{s}^{\leq l-1})\}_{l=1}^{L}$ are products of independent distributions, and the lowest level marginal $q^{\mathrm{NN}}(\boldsymbol{s}^0)$ is the Boltzmann distribution with only $2^{2 \times 2}$ states.

Figure 3 left shows the relative estimation error $(\hat{F} - F)/|F|$ where $F$ is the ground-truth free energy, computed analytically (Onsager, 1944). Each error bar shows one standard deviation of the statistical error. We observe that MLMC-HB already performs poorly for $N \geq 32$, exhibiting strong biases.[5] The other four methods achieve compatible (unbiased) estimation up to $N \leq 32$, with Cluster-AIS and our RiGCS outperforming VAN and HAN in terms of the variance. Furthermore, for $N \geq 64$, the vanilla VAN cannot be trained because its wall-clock training time exceeds several weeks, and the variance for HAN is much larger compared to RiGCS. For $N = 128$, only Cluster-AIS and RiGCS achieve results compatible with the ground truth free energy, while HAN gives highly biased estimates incompatible with the ground truth. These results demonstrate the superiority of our RiGCS over existing generative models for estimating thermodynamic observables.

## 4.2 QUALITY MEASURES FOR GENERAL OBSERVABLE ESTIMATION

We further compare the samplers using a recently proposed Effective Mode-Dropping Measure (EMDM) (Nicoli et al., 2023) and the commonly used Effective Sample Size (ESS) as indicators of bias and variance, respectively, for general (non-thermodynamic) observable estimation. The EMDM is defined as $\bar{w} = \mathbb{E}_{\tilde{q}_\theta}[w(\boldsymbol{s})] \in [0, 1]$, where $w(\boldsymbol{s}) = \frac{p(\boldsymbol{s})}{q_\theta(\boldsymbol{s})}$, and $\tilde{q}_\theta$ is the *renormalized density* of $q_\theta(\boldsymbol{s})$ *with effective support*—i.e., the support excluding the low density areas where no sample appears with high probability. [6] Nicoli et al. (2023) showed that the bias of the importance-weighted estimators for general observables can be bounded with EMDM, and $\bar{w} = 1$ indicates that no effective mode-dropping occurs and thus the estimator is unbiased. ESS (per sample), defined as $\mathrm{ESS} = \frac{1}{\mathbb{E}_{q_\theta}[w(\boldsymbol{s})^2]} \in [0, 1]$, is known to be inversely proportional to the variance of general unbiased estimators, and $\mathrm{ESS} = 1$ implies that $q_\theta(\boldsymbol{s}) = p(\boldsymbol{s})$.

Figure 4 (left) shows the $\bar{w}$ (EMDM) for RiGCS, VAN and HAN. We recall that the vanilla VAN can be trained only up to $N = 32$, as explained above. For $N \leq 64$, the EMDMs of HAN and RiGCS are

---

[5]The results for $N \geq 64$ are out of the range.

[6]The threshold for the "low density area" depends on the number $M$ of samples, and $\tilde{q}_\theta(\boldsymbol{s}) = q_\theta(\boldsymbol{s})$ for $M \to \infty$.

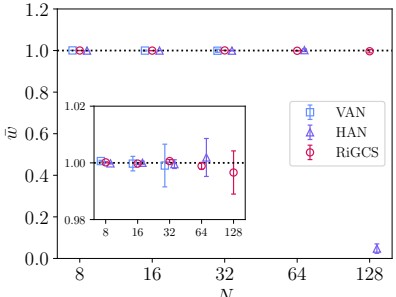 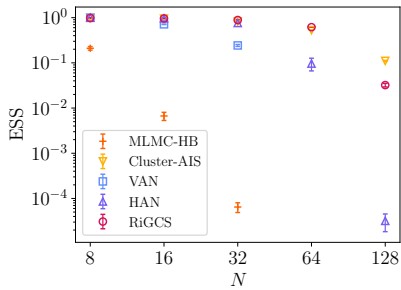

Figure 4: EMDM (left) and ESS (right). The vanilla VAN cannot be trained for $N \geq 64$ in reasonable time. The inset in the left plot zooms around the region of $\bar{w} \approx 1.0$, showing that RiGCS does not suffer from mode dropping even for large $N = 128$. Similarly, the ESS plot (right) shows that RiGCS achieves the closest performance to Cluster-AIS for large systems.

compatible with EMDM $\approx 1$, indicating no effective mode-dropping. However, for $N = 128$, the EMDM of HAN drops significantly, implying that the model is affected by effective mode-dropping. This result is consistent with the biased free energy estimation by HAN observed in Figure 3 (left). RiGCS shows no evidence of mode-dropping for $N = 128$, highlighting its robustness in accurately modeling the target distribution and avoiding mode collapse in high dimensions. Figure 4 (right) shows that RiGCS outperforms all baselines except the cluster method, a state-of-the-art approach for general observable estimation. Note that for $N = 128$, RiGCS improves the ESS of HAN by several orders of magnitude, making it the only generative model with *non-vanishing* ESS.

**Simulation Costs**   Generative modeling approaches like VAN, HAN, and RiGCS require training, with wall-clock times for a $N = 64$ lattice of approximately 60 days, 2.8 hours, and 3.8 hours, respectively. Sequential training with model transfer (introduced in Section 3.3) offers a significant advantage over random initialization, as shown in Figure 3 (right). Sampling costs for generating 100 samples with MLMC-HB, VAN, HAN, and RiGCS are approximately 14, 27, 0.2, and 0.4 seconds, respectively.

Further investigation including additional analysis, ablation study with transformer architectures, and benchmarking for different physical theories are provided in Appendix H.

## 5   CONCLUSIONS

Critical phenomena such as phase transitions are of high relevance in many fields of physics, where Renormalization Group Theory (RGT) plays a central role for theoretical analysis. Insights from RGT were also used for improving tools for numerical analysis, leading to MultiLevel Monte Carlo (MLMC) methods based on the emerging *scale invariance at criticality* (SIC). In this paper, we further enhanced such tools by leveraging machine learning techniques. Specifically, we introduced Renormalization-informed Generative Critical Sampler (RiGCS), a multilevel sampling algorithm where conditional generative models, with appropriate size of receptive fields, are substituted to naive nearest-neighbor heat bath conditional samplers. This modification allows for capturing long-range and higher-order interactions that exist under slight violation of SIC, making RiGCS the new state-of-the-art generative samplers. Furthermore, we also introduced the *sequential training with model transfer*, which was again inspired by SIC and significantly reduced the training cost. Although many previous works incorporated general domain knowledge, e.g., invariances, equivariances, and preservation laws, for machine learning model design, this work is one of the few applications where the knowledge of critical phenomena, i.e., SIC, is incorporated directly in both the architecture design and the training procedure. We see this work as a first step toward developing specialized machine learning methods for critical regimes, paving the way for more efficient algorithms. Future research directions include applying RiGCS to other physical models, such as Potts models and lattice gauge theories, as well as exploring hybrid approaches that combine RiGCS with related methods, such as HAN.

ACKNOWLEDGMENTS

The authors thank Piotr Białas, Piotr Korcyl, Tomasz Stebel, and Dawid Zapolski for sharing their code to reproduce the results with their Hierarchical Autoregressive Networks (HAN), and Andrea Bulgarelli for sharing a highly optimized implementation of the cluster algorithm, which was used to obtain the ground truth reference values. The authors also thank Christopher Anders and Pan Kessel for fruitful discussions and inspiration. This work was supported by the German Ministry for Education and Research (BMBF) under the grant BIFOLD24B, the Deutsche Forschungsgemeinschaft (DFG, German Research Foundation) as part of the CRC 1639 NuMeriQS – project no. 511713970, the SFT Scientific Initiative of INFN, the Simons Foundation grant 994300 (Simons Collaboration on Confinement and QCD Strings), and the European Union's HORIZON MSCA Doctoral Networks program project AQTIVATE (101072344).

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

## A  EXTENDED RELATED WORK

Renormalization Group Theory (RGT) has significantly impacted the study of statistical systems, especially in analyzing critical phenomena and phase transitions. The pioneering works by Wilson, Kogut, and Kadanoff laid the foundational principles of RG theory (Wilson, 1971; Wilson & Kogut, 1974; Kadanoff, 1966). Subsequent advancements have expanded the application of RGT-inspired methods to disordered systems, random ferromagnetic chains, and Monte Carlo simulations (Aharony, 1973; Fisher, 1973; Swendsen, 1979; Derrida, 1980; Butera & Comi, 2002). These developments have enriched the understanding of critical phenomena and further established the applications of RGT techniques in both theoretical and computational contexts. For further insights into RG we refer the reader to Appendix C.

In computational physics, substantial research has focused on RGT-inspired sampling methods for lattice simulations. For example, early studies on the Ising model (Schmidt, 1983; Faas & Hilhorst, 1986) achieved notable success in simulating small systems in one dimension, yet faced limitations as the lattice size increased. More recently, Jansen et al. (2020) introduced a new theoretical framework for low variance estimation, showing promising results for one-dimensional quantum systems. In lattice gauge theory, the application of RGT concepts led to the development of several algorithms, including multigrid (Cahill & Kogut, 1982; Goodman & Sokal, 1986; Hulsebos & Hockney, 1989), multiscale thermalization techniques (Endres et al., 2015), and decimation maps (Matsumoto et al., 2023).

In recent years, RGT-inspired approaches have been combined with machine learning to develop more scalable neural samplers. Notable examples include applications in $U(1)$ (Finkenrath, 2024) and $SU(3)$ gauge theories (Abbott et al., 2024), where a renormalization group (RG) scheme has been combined with normalizing flows. Similarly to our approach, Białas et al. (2022) proposed a Hierarchical Autoregressive Network (HAN) for sampling configurations of the 2D Ising model. This latter leverages a recursive domain decomposition technique (Cè et al., 2016), in which different regions of the configuration are sampled in parallel using the same autoregressive network, thus replacing the traditional scaling with the system's linear extent $L$. Białas et al. (2022) demonstrated the effectiveness of HAN on the two-dimensional Ising model, simulating lattices up to the size of $128 \times 128$. However, this method has several limitations, which are discussed in Section 4 and Appendix H in great detail. On a side note, we emphasize that our multilevel approach and the domain decomposition proposed in Białas et al. (2022) are not mutually exclusive. In fact, these methods could in principle be combined leading to more powerful sampling protocols. We defer this investigation to future work.

Besides the development of enhancing sampling methods, other recent works have leveraged the idea of RG in different ways. Li & Wang (2018) focused on *neural network renormalization group*, investigating the capability of neural networks to perform hierarchical feature extraction and hierarchical transformations. To this end, they used bijective transformations to learn hierarchical maps to automatically identify mutually independent collective variables. While inspired by RGT, this approach does not use multilevel sampling, and does not observe favorable scaling—in the paper only results on rather small lattices up to $16 \times 16$ are shown. Koch-Janusz & Ringel (2018) proposed a machine learning approach to identify the relevant degrees of freedom and extract Ising critical exponents in one and two-dimensional systems. Efthymiou et al. (2019) leveraged the idea of image super-resolution and trained convolutional neural networks that invert real-space renormalization decimations. The authors showed that it is possible to predict thermodynamic quantities for lattice sizes larger than those used in training. Lenggenhager et al. (2020) drew a connection between real-space RG and real-space mutual information. From an information-theoretic standpoint, they investigated the information loss at arbitrary coarse graining of the lattices through the lens of RG. A more recent study by Marchand et al. (2023) introduced the Wavelet-Conditional Renormalization Group (WCRG), where fast wavelet transforms are used to build an RG transformation across scales. While similar in spirit, their approach is substantially different from ours. Specifically, they train the model by using a contrastive divergence loss, which requires a lot of training samples drawn from the target distributions. Furthermore, Hu et al. (2020) used the neural network renormalization group (Li & Wang, 2018) as a universal approach to design generic Exact Holographic Maps (EHMs) for interacting field theories. Chung & Kao (2021) used Restricted Boltzmann Machines (RBMs) to learn a valid real-space RG transformation without prior knowledge of the physical system, establishing a solid connection between the RG transformation in physics and the statistical learning theory. Ron

et al. (2021) and Bachtis et al. (2022), instead, used modified block-spin transformations—to improve convergence in the Monte Carlo (MC) renormalization group trajectory—and inverse RG transformations, respectively, to extract critical exponents of a given physical theory.

Recently, a so-called Machine-Learning Renormalization Group (MLRG) algorithm has been developed to explore and analyze many-body lattice systems in statistical physics (Hou & You, 2023). In a recent work by Di Sante et al. (2022), the authors proposed a data-driven dimensionality reduction and used a Neural Ordinary Differential Equation (NODE) solver in a low-dimensional latent space to efficiently learn the functional RG dynamics. The authors showed promising results in the context of the Hubbard model.

Lin et al. (2017) pointed out that convolutional neural networks in supervised learning tasks can act as a "coarse-graining" procedure, isolating relevant macroscopic features from irrelevant microscopic noise. In recent years, RG-inspired machine learning applications have emerged in the context of variational inference (Ahn et al., 2018), regularization techniques (Wang et al., 2024), transfer learning (Redman et al., 2022), and multi-scale semantic manipulation of images (Hu et al., 2022). While all these related works leverage the concept of RG in different ways—such as for extracting critical exponents, or interpreting RG as coarse-graining procedures in machine learning—they suffer from a few shortcomings when it comes to efficient sampling. First, they often lack the access to a tractable probability density, which is required for importance sampling or neural MC to obtain unbiased estimates. Second, they do not allow for data-free training of a neural sampler as well as rapid and effective sampling at multiple scales—as RiGCS does.

RG ideas have also been used in the context of Tensor Networks (TNs) to construct an emergent scale invariant description for critical systems. TNs describe the wave function, or the partition function, of a system as a contraction of a network of smaller tensors. This approach was shown to be efficient as long as the entanglement in the system is moderate (Bridgeman & Chubb, 2017). Blocking tensors together and coarse graining the system allow for (numerically) obtaining a description of the system at a larger length scale. Prominent algorithms for coarse graining the partition function of a critical system are the Tensor Network Renormalization (TNR) group (Evenbly & Vidal, 2015) and loop TNR (Yang et al., 2017) for square lattices, as well as Graph-Independent Local Truncations (GILT) for arbitrary graphs (Hauru et al., 2018). The Multi-scale Entanglement Renormalization Ansatz (MERA) (Vidal, 2008; Evenbly & Vidal, 2014) leverages the hierarchical structure of RG to efficiently represent quantum states for critical systems in 1+1 dimensions that are described by an underlying conformal field theory. Generalizing this idea, and understanding holographic duality as a generalization of the RG flow, Qi (2013) and Lee & Qi (2016) proposed an exact holographic mapping, which is a one-to-one unitary mapping between boundary and bulk degrees of freedom. Unlike MC methods, TN approaches enable the direct computation of the expectation values of observables, as they provide an approximation to the wave function or the partition function of a system. However, although the numerical algorithms for TN methods, which allow for recovering exact scale invariance at the critical point, scale polinomially with respect to both system size and tensor size, the computational cost remains challenging due to a large degree in the tensor size $\chi$—the leading order costs of TNR and MERA are $\mathcal{O}(\chi^6)$, and $\mathcal{O}(\chi^9)$, respectively.

Lastly, Cotler and Rezchikov have recently uncovered intriguing connections between RG theory and optimal transport (Cotler & Rezchikov, 2023a) and diffusion models (Cotler & Rezchikov, 2023b), highlighting promising new directions for further investigation.

# B   MARKOV CHAIN MONTE CARLO AND CLUSTER METHODS

## B.1   MARKOV CHAIN MONTE CARLO (MCMC) SAMPLING

MCMC methods allow for producing a sequence of configurations $\{s^{(1)}, s^{(2)}, \dots\}$ through a Markov chain, following an *target distribution* $p(s) = \frac{1}{Z}\widetilde{p}(s)$ for which the partition function $Z$ is unknown. To this end, given the current configuration $s^{(m)}$, a new configuration $s'$ is proposed, which is either accepted or rejected. If accepted, it becomes the next configuration, i.e., $s^{(m+1)} = s'$, while if rejected, the configuration stays, i.e., $s^{(m+1)} = s^{(m)}$. To ensure that the produced configurations follow the target distribution, the transition probability $T(s \rightarrow s')$ from configuration $s$ to

$s'$ has to fulfill

$$\sum_s \widetilde{p}(s)T(s \to s') = \widetilde{p}(s') = \sum_s \widetilde{p}(s')T(s' \to s). \tag{14}$$

One way to ensure the condition above is by making the transition probability satisfy the *detailed balance condition*:

$$\widetilde{p}(s)T(s \to s') = \widetilde{p}(s')T(s' \to s). \tag{15}$$

Together with the property of *ergodicity*—i.e., existence of at least one Markov chain connecting any pair of configurations with positive transition probability—the detailed balance condition ensures that the configurations sampled by the Markov process follow the target distribution $p(s)$ after thermalization.

Once a sufficient number of configurations are sampled, the expectation values of physical observables $\mathcal{O}(s)$, e.g., energy, magnetization, and correlation functions, can be estimated by averaging over the configurations

$$\langle \mathcal{O} \rangle \approx \frac{1}{M - \underline{m}} \sum_{m=\underline{m}+1}^{M} \mathcal{O}(s^{(m)}), \tag{16}$$

where $M$ is the total number of sampled configurations, and $\underline{m}(< M)$ is the number of *burn-in* steps for thermalization. Due to the dependence of each sample configuration on the previous one in the Markov chain, MCMC samples are inherently correlated. This correlation is quantified by the *autocorrelation time* $\tau$, which characterizes the degree and persistence of correlation between samples. A long autocorrelation time can significantly degrade the accuracy of the Monte Carlo estimator (16), because it makes the Effective Sample Size (ESS) much smaller than the generated number of samples.

Crucially, as the system approaches the critical point, the autocorrelation time $\tau$ of thermodynamic properties diverges, following a power law in the correlation length $\xi$:

$$\tau \propto \xi^z, \tag{17}$$

where $z$ is known as the *dynamical critical exponent*. Due to scale invariance at criticality (SIC), the correlation length diverges ($\xi \to \infty$) as the system approaches the critical point. This means that the autocorrelation time diverges, as Eq. (17) implies, leading to a phenomenon called *critical slowing down* (Wolff, 1990). For finite hypercubic lattices, the correlation length in lattice units is bounded by the extent $N$ of the lattice in each dimension, hence

$$\tau \propto N^z, \tag{18}$$

as one approaches the critical point. From Eq. (18), it is clear that, depending on the critical exponent $z$, sampling configurations with high ESS at criticality becomes increasingly challenging as the lattice size $N$ increases. For local update protocols, as for example Metropolis-Hastings, one typically obtains values of $z \approx 2$ (Wolff, 1990). This increase in autocorrelation necessitates a larger number of samples to achieve accurate statistical estimates, thereby raising the computational cost (Schaefer et al., 2011).

## B.2 CLUSTER ALGORITHMS

In contrast to the algorithms relying on local updates, cluster algorithms can yield the dynamical critical exponents close to zero, thus avoiding critical slowing down. In the following, we briefly review two cluster algorithms for efficient sampling close to the critical point in the Ising model,

$$p(s) = \tfrac{1}{Z} \exp\left(-\beta \sum_{\langle v,v' \rangle} J s_v s_{v'}\right), \tag{19}$$

where $s \in \{-1, 1\}^V$ is the spin configuration, and $\sum_{\langle \cdot, \cdot \rangle}$ takes the sum over all nearest neighbor pairs.

### B.2.1 SWENDSEN-WANG ALGORITHM

The basic principle of the Swendsen-Wang algorithm is to flip entire *clusters* of spins instead of a single one (Swendsen & Wang, 1987). To this end, a given spin configuration is divided into clusters by forming *bonds* between spins. A cluster then consists of all spins connected directly or indirectly via bonds. Subsequently, all the spins belonging to a cluster are flipped collectively. More specifically, given the current configuration $\boldsymbol{s}^{(m)} \in \{-1, 1\}^V$, the algorithm proceeds as follows:

1. Inspect all nearest-neighbor pairs $\langle v, v' \rangle$ in $\boldsymbol{s}^{(m)}$. If $s_v^{(m)} = s_{v'}^{(m)}$, a bond is formed between the pair with probability $1 - \exp(-2\beta J)$. Otherwise, no bond is formed.

2. Identify all clusters, i.e., all sets of spins connected either directly or indirectly by the bonds.

3. Flip all spins within each cluster collectively with a certain probability $P^{\text{flip}}$, resulting in a new spin configuration $\boldsymbol{s}'$.

4. Remove all bonds and repeat the steps for the new spin configuration $\boldsymbol{s}^{(m+1)} = \boldsymbol{s}'$.

In step 3, if $P^{\text{flip}}$ is chosen to be close to zero, the new configuration $\boldsymbol{s}'$ will, in general, be too similar to $\boldsymbol{s}$. In contrast, choosing $P^{\text{flip}} = 1$ will result in a full inversion of the configuration $\boldsymbol{s}^{(m)}$, which does not change the energy at all. As both extremal cases do not produce sensible new configurations, $P^{\text{flip}}$ is typically set to $1/2$.

The Swendsen-Wang algorithm can be viewed as a data augmentation method (Higdon, 1998; Kasteleyn & Fortuin, 1969; Fortuin & Kasteleyn, 1972), where the configuration space of the Ising model is extended by introducing auxiliary bond variables $u_{\langle v, v' \rangle} \in [0, e^{2\beta J}]$ for nearest-neighbor pairs. Specifically, we assume that $u_{\langle v, v' \rangle}$ follows the conditional uniform distribution:

$$p(u_{\langle v,v' \rangle} | s_v, s_{v'}) = \exp(-\beta J(1 + s_v s_{v'})) \cdot \mathbb{1}(u_{\langle v,v' \rangle} \in [0, \exp(\beta J(1 + s_v s_{v'}))]), \qquad (20)$$

where $\mathbb{1}(\cdot)$ is the identity function equal to one if the event is true and zero otherwise. This gives the entire joint distribution as

$$p(\{u_{\langle v,v' \rangle}\}, \boldsymbol{s}) = p(u_{\langle v,v' \rangle} | \boldsymbol{s}) p(\boldsymbol{s}) \propto \prod_{\langle v,v' \rangle} \mathbb{1}(u_{\langle v,v' \rangle} \in [0, \exp(\beta J(1 + s_v s_{v'}))]). \qquad (21)$$

Noting that

$$\exp(\beta J(1 + s_v s_{v'})) = \begin{cases} 0 & \text{if } s_v \neq s_{v'}, \\ \exp(2\beta J) > 1 & \text{if } s_v = s_{v'}, \end{cases}$$

Eq. (21) gives the following conditional distribution:

$$p(s_v, s_{v'} | u_{\langle v,v' \rangle}) = \begin{cases} \frac{1}{2} & \text{for } (s_v, s_{v'}) \in \{\{-1, -1\}, \{1, 1\}\} & \text{if } u_{\langle v,v' \rangle} > 1, \\ 0 & \text{for } (s_v, s_{v'}) \in \{\{-1, 1\}, \{1, -1\}\} & \text{if } u_{\langle v,v' \rangle} > 1, \\ \frac{1}{4} & \text{for } (s_v, s_{v'}) \in \{-1, 1\}^2 & \text{if } u_{\langle v,v' \rangle} \leq 1, \end{cases} \qquad (22)$$

Since Eq. (22) depends on $u_{\langle v,v' \rangle}$ only through whether it is larger than $1$ or not, and our goal is to sample configurations $\boldsymbol{s}$, we can replace the continuous random variable $u_{\langle v,v' \rangle}$ with the binary random variable $b_{\langle v,v' \rangle} \in \{0, 1\}$ such that the event $b_{\langle v,v' \rangle} = 1$ corresponds to the event $u_{\langle v,v' \rangle} > 1$. From Eqs. (20) and (22), we have

$$\begin{aligned} p(b_{\langle v,v' \rangle} = 1 | s_v, s_{v'}) = p(u_{\langle v,v' \rangle} > 1 | s_v, s_{v'}) &= \frac{\exp(\beta J(1 + s_v s_{v'})) - 1}{\exp(\beta J(1 + s_v s_{v'}))} \\ &= (1 - \exp(-2\beta J)) \cdot \mathbb{1}(s_v = s_{v'}), \qquad (23) \end{aligned}$$

and

$$p(s_v, s_{v'} | b_{\langle v,v' \rangle}) = \begin{cases} \frac{1}{2} & \text{for } (s_v, s_{v'}) \in \{\{-1, -1\}, \{1, 1\}\} & \text{if } b_{\langle v,v' \rangle} = 1, \\ 0 & \text{for } (s_v, s_{v'}) \in \{\{-1, 1\}, \{1, -1\}\} & \text{if } b_{\langle v,v' \rangle} = 1, \\ \frac{1}{4} & \text{for } (s_v, s_{v'}) \in \{-1, 1\}^2 & \text{if } b_{\langle v,v' \rangle} = 0, \end{cases} \qquad (24)$$

respectively. Notably, the spins between the neighboring pairs with a bond $b_{\langle v,v' \rangle} = 1$ must be the same, and those without bond $b_{\langle v,v' \rangle} = 0$ have *no interactions* in the conditional (24). This allows for *cluster-wise* independent sampling without rejection step. Swendsen-Wang algorithm thus performs Gibbs sampling with Eq. (23) (Step 1) and Eq.(24) (Step 3) iteratively to obtain a Markov chain of spin configurations $\{\boldsymbol{s}^{(m)}\}$.

The effectiveness of the Swendsen-Wang algorithm can be understood by the fact that flipping large clusters allows for efficiently destroying the long-range correlations emerging close to the critical point. For $D > 2$, the Swendsen-Wang algorithm becomes less capable, as the majority of the formed clusters tend to be small, with only a few large ones being generated.

### B.2.2 WOLFF ALGORITHM

The Wolff algorithm (Wolff, 1989a) is a single-cluster variant of the Swendsen-Wang algorithm. Instead of dividing the entire configuration into clusters and flipping each of them, the Wolff algorithm only forms a single cluster and collectively flips the spins inside this cluster. Given the current configuration $\boldsymbol{s}^{(m)} \in \{-1, 1\}^V$, the Wolff algorithm proceeds with the following steps:

1. Randomly choose a site $v \in \{1, \dots, V\}$.
2. Form bonds, analogous to the Swendsen-Wang algorithm, with probability $1 - \exp(-2\beta J)$ with all nearest neighbors $\{v'\}$ such that $s_v = s_{v'}$.
3. For each of the bonded sites $\{v'\}$, form bonds with its respective neighbors that have not been bonded, according to Step 2.
4. Repeat Step 3 iteratively until no more bonds can be formed.
5. Flip all spins within the cluster and obtain a spin configuration $\boldsymbol{s}'$.
6. Remove all bonds and repeat the steps for the new spin configuration $\boldsymbol{s}^{(m+1)} = \boldsymbol{s}'$.

Note that compared to the Swendsen-Wang algorithm, the cluster is flipped with probability $P^{\text{flip}} = 1$. If the cluster formed by the Wolff algorithm is large, the long-range correlations are broken up essentially as effectively as in the Swendsen-Wang algorithm with a smaller computational cost—because the Wolff algorithm focuses only on a single cluster. If the cluster formed by the Wolff algorithm is small, the configuration does not change significantly, however, at the same time, the computational cost is also small. Thus, the Wolff algorithm turns out to be even more efficient in decreasing the dynamical critical exponent $z$, compared to the Swendsen-Wang approach (Swendsen & Wang, 1987; Wolff, 1989b).

In our experiments in Section 4 and Appendix H, we used the Wolff algorithm as the state-of-the-art *Cluster* method, which is more efficient than the Swendsen-Wang algorithm.

## C  RENORMALIZATION GROUP

The Renormalization Group (RG) (Wilson, 1971; Cardy, 1996) is a powerful framework in theoretical physics for studying the behavior of systems as they are progressively coarse-grained to larger length scales. During this process, microscopic degrees of freedom are systematically marginalized, generating a flow in parameter space known as the RG flow. More formally, given a Hamiltonian $H$ describing the system at a given length-scale, one can define an RG transformation $\mathcal{R}_\lambda$ as

$$H' = \mathcal{R}_\lambda [H].$$

Namely, the operator $\mathcal{R}_\lambda$ changes the scale of the system and yields a *renormalized Hamiltonian $H'$* describing the system at a larger length scale, indexed by $\lambda$, with less degrees of freedom. For $\mathcal{R}_\lambda$ to be a proper RG transformation, it has to fulfill the semi-group property, i.e., there exists a neutral element $\mathcal{R}_0$ that does not change the scale and the composition of two transformations to different length scales $\lambda$ and $\lambda'$ has to fulfill $\mathcal{R}_{\lambda'} \circ \mathcal{R}_\lambda = \mathcal{R}_{\lambda'+\lambda}$. This transformation generates a flow on the space of Hamiltonians that can yield crucial insights into the macroscopic properties of physical systems. In particular, the critical points correspond to the *fixed points $H^*$* satisfying

$$H^* = \mathcal{R}_\lambda [H^*] \tag{25}$$

in the RG flow, as the system exhibits scale invariance at criticality (SIC).

Close to the critical point, one can approximate the Hamiltonian of the system as $H = H^* + \delta H$, where $\delta H$ is a small perturbation. Expanding the RG transformation around the fixed point, one finds

$$\mathcal{R}_\lambda \left[ H^* + \delta H \right] = H^* + \mathcal{L} \left[ H^* \right] \delta H + \mathcal{O} \left( \delta H^2 \right) \approx H^* + \delta H', \tag{26}$$

where $\delta H' = \mathcal{L} \left[ H^* \right] \delta H$. Applying the transformation $\mathcal{R}_\lambda$ for $\tau$-times, we find in leading order that

$$\mathcal{R}_{\tau \times \lambda} \left[ H^* + \delta H \right] \approx H^* + \mathcal{L} \left[ H^* \right]^\tau \delta H. \tag{27}$$

Expanding $\delta H$ in the eigenoperators $\{M_j\}$ of $\mathcal{L} \left[ H^* \right]$, one finds that the leading order correction can be expressed as

$$\mathcal{L} \left[ H^* \right]^\tau \delta H = \sum_j c_j e_j^\tau M_j, \tag{28}$$

where $\{c_j\}$ are the expansion coefficients, and $\{e_j\}$ are the eigenvalues corresponding to the eigenoperators $\{M_j\}$.

For a large $\tau$, or equivalently at large length scales, one observes that the eigenvalues $\{e_j\}$ determine the behavior of the system: the renormalized Hamiltonian tends to be dominated by the eigenoperators with larger eigenvalues. We say that the eigenoperator $M_j$ is *relevant*, *marginally relevant*, or *irrelevant* if the corresponding eigenvalues are $e_j > 1$, $e_j = 1$, or $e_j < 1$, respectively. Then, Eq. (28) implies that the relevant and marginally relevant operators determine the macroscopic behavior of the system. Thus, close to the critical point, the systems sharing the same (marginally) relevant operators will exhibit the same behavior at macroscopic scales, regardless of the microscopic degrees of freedom. This gives rise to the notion of universality classes, i.e., physical systems showing the same scaling behavior at criticality described by typically a few critical exponents, despite being microscopically different (Cardy, 1996). Thus, information about a system's behavior close to criticality can be obtained by studying another model within the same universality class. For example, the Ising model, originally developed to describe phase transitions in ferromagnetic systems, can also be used to study the liquid-gas transition, superfluids, and the Higgs mechanism (Wilson, 1971; Wilson & Kogut, 1974).

A simple example of an RG transformation is the Kadanoff block spin transformation, which will be illustrated below. The partition function of the Ising Hamiltonian is given by

$$Z = \sum_s \exp \left( -\beta H(s) \right) = \sum_s \exp \left( -\beta J \, \boldsymbol{s} \boldsymbol{I}^{\mathrm{NN}} \boldsymbol{s}^\top \right), \tag{29}$$

where $\boldsymbol{I}^{\mathrm{NN}}$ is the nearest neighbor matrix (of the appropriate size depending on the context) defined as

$$(\boldsymbol{I}^{\mathrm{NN}})_{v, v'} = \begin{cases} 1 & \text{if } (v, v') \text{ are } 2 \cdot D \text{ nearest neighbor pairs,} \\ 0 & \text{otherwise.} \end{cases} \tag{30}$$

For $D = 1$ (see the top row in Figure 5), this can be rewritten (Maris & Kadanoff, 1978) as

$$Z = \sum_{s_1, s_3, s_5, \ldots} \left( \sum_{s_2, s_4, s_6, \ldots} e^{K(s_1 s_2 + s_2 s_3)} e^{K(s_3 s_4 + s_4 s_5)} \cdots \right) \tag{31}$$

$$= \sum_{s_1, s_3, s_5, \ldots} \left[ e^{K(s_1 + s_3)} + e^{-K(s_1 + s_3)} \right] \left[ e^{K(s_3 + s_5)} + e^{-K(s_3 + s_5)} \right] \cdots, \tag{32}$$

where $K = -\beta J$. Here, we have separated the sum over the even and odd spins in Eq. (31), and explicitly performed the sum over the even spins in Eq. (32). Since $s_v \in \{-1, 1\}$, it holds that

$$e^{K(s_1 + s_3)} + e^{-K(s_1 + s_3)} = f(K) e^{K' s_1 s_3}, \tag{33}$$

where

$$f(K) = 2 \sqrt{\cosh(2K)}, \qquad\qquad K' = \ln(\cosh(2K))/2. \tag{34}$$

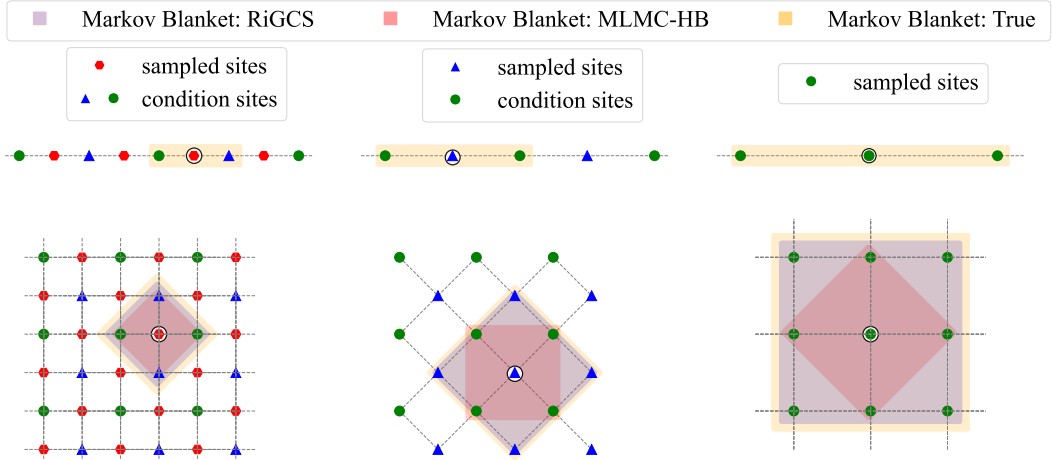

Figure 5: Site partitionings based on the block-spin transformation (Kadanoff, 1966). This figure is similar to, yet more accurate than, Figure 1. Specifically, the only difference is in the true Markov blanket (yellow shadow) of the second finest level in the 2D case (bottom row in the middle). While Figure 1 is for explaining that the true Markov blankets of the intermediate levels—which in general are *unknown*—*possibly* covers the entire lattice, this figure faithfully shows the Markov blanket for the true renormalized Hamiltonian (36) at the second finest level.

Substituting Eq. (33) into Eq. (32), one finds

$$Z = f(K)^{N/2} \sum_{s_1, s_3, s_5, \ldots} \exp(K' s^{\leq L-1} I^{\mathrm{NN}} (s^{\leq L-1})^\top)$$

$$= f(K)^{N/2} \sum_{s^{\leq L-1}} \exp\left(-\beta \widetilde{H}_{L-1} (s^{\leq L-1}; -K'/\beta)\right),\tag{35}$$

where

$$\widetilde{H}_l(s^{\leq l}; J) = -J s^{\leq l} I^{\mathrm{NN}} (s^{\leq l})^\top$$

is the nearest-neighbor renormalized Hamiltonian. This demonstrates that the partition function of the system on the fine lattice is related to the one on a coarser lattice described by the same type of Hamiltonian with $K'$ different from $K = -\beta J$. Moreover, looking at Eq. (34), the only fixed points $K^*$ in the recursion relation for the renormalized couplings are the trivial ones, i.e., $K^* = 0$, corresponding to the temperature $T = \beta^{-1} \to \infty$, where the system is in the paramagnetic phase, and $K^* = \infty$, corresponding to $T = 0$, where the system is in the ferromagnetic phase.

For $D = 2$ (see the bottom row in Figure 5), one can follow a similar approach by marginalizing at each iteration over the even (or odd) degrees of freedom in a "checker board" pattern (see also Figure 6). After a single step of the procedure, one obtains the following partition function (Maris & Kadanoff, 1978):

$$Z = f(K)^{N/2} \sum_{s^{\leq L-1}} \exp\left(K_1 \sum_{\langle ij \rangle} s_i s_j + K_2 \sum_{\langle\langle ij \rangle\rangle} s_i s_j + K_3 \sum_{\square} s_i s_j s_r s_t\right),\tag{36}$$

where $\langle ij \rangle$ corresponds to the nearest-neighbors on the lattice after summing over one sublattice, $\langle\langle ij \rangle\rangle$ to the spins on next nearest-neighbor sites, $\square$ indicates the spins on a plaquette of the coarser lattice, and

$$K_1 = \frac{1}{4} \ln\left(\cosh\left(4K\right)\right), \ \ K_2 = \frac{1}{8} \ln\left(\cosh\left(4K\right)\right), \ \ K_3 = \frac{1}{8} \ln\left(\cosh\left(4K\right)\right) - \frac{1}{2} \ln\left(\cosh\left(2K\right)\right).$$

Note that in this case, the partition function is not given by the exponential of the same type of Hamiltonian as the original model just with different parameters on a coarser lattice. This is illustrated in Figure 5 as the "true Markov blanket" (yellow shadow) of the second finest level lattice in

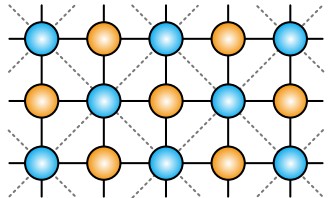

Figure 6: Illustration of the Kadanoff block spin method on a square lattice, the spheres indicate the spins and the solid black lines the original lattice. After summing over the configurations of the orange spins, one obtains an renormalized Hamiltonian for the blue spins on a square lattice that is tilted by 45° relative to the original lattice (indicated by the grey dashed lines). The resulting Hamiltonian on the dashed lattice contains nearest-neighbor interactions, interactions of all four spins along a square as well as next nearest-neighbor interactions along the diagonal of the squares.

the 2D case (the middle figure in the bottom row). Continuing this procedure, one would generate various long-range and multi-body interactions in the renormalized Hamiltonian, which is captured in the Hamiltonian in Eq. (9).

A practical example of RG flow in machine learning is the application of CNNs to a classification problem (Lin et al., 2017). CNNs perform a form of coarse-graining, where successive convolutional layers progressively filter out microscopic noise (irrelevant operators) and isolate high-level features (relevant operators) essential for distinguishing the target classes. The latter example can be tested by training a CNN to classify the phase (ferromagnetic or paramagnetic) of the Ising model. As shown in Carrasquilla & Melko (2017), the output of such a CNN is strongly correlated with the magnetization, indicating that both neural networks and the RG flow capture the same key parameter—magnetization—as a relevant feature to characterize the phase transition in the Ising model. Interestingly, a similar behavior was observed in Alexandrou et al. (2020), where the latent representation of an autoencoder trained on Ising configurations was studied.

## D MULTILEVEL MONTE CARLO WITH HEAT BATH (MLMC-HB) ALGORITHM

With the site partitioning (see Figure 1) based on the block-spin transformations (Kadanoff, 1966), MLMC-HB performs ancestral sampling, according to Eq. (8), i.e.,

$$q^{\text{NN}}(\boldsymbol{s}) = p(\boldsymbol{s}^L|\boldsymbol{s}^{\leq L-1}) \left( \prod_{l=1}^{L-1} q^{\text{NN}}(\boldsymbol{s}^l|\boldsymbol{s}^{\leq l-1}) \right) q^{\text{NN}}(\boldsymbol{s}^0),$$

which approximates the target distribution (7):

$$p(\boldsymbol{s}) = \left( \prod_{l=1}^{L} p(\boldsymbol{s}^l|\boldsymbol{s}^{\leq l-1}) \right) p(\boldsymbol{s}^0).$$

Here,

$$q^{\text{NN}}(\boldsymbol{s}^{\leq l}) = \left( \prod_{l'=1}^{l-1} q^{\text{NN}}(\boldsymbol{s}^{l'}|\boldsymbol{s}^{\leq l'-1}) \right) q^{\text{NN}}(\boldsymbol{s}^0)$$

for $l = 0, \ldots, L-1$ approximates the true marginal distribution (4), i.e.,

$$p(\boldsymbol{s}^{\leq l}) = \int p(\boldsymbol{s}) \mathcal{D}[\boldsymbol{s}^{>l}] \equiv \frac{1}{Z_l} e^{-\beta H_l(\boldsymbol{s}^{\leq l})},$$

with nearest-neighbor (NN) interaction Hamiltonians, i.e.,

$$q^{\text{NN}}(\boldsymbol{s}^{\leq l}) = \frac{1}{\widetilde{Z}_l} e^{-\beta \widetilde{H}_l(\boldsymbol{s}^{\leq l})} \quad \text{with} \quad \widetilde{H}_l(\boldsymbol{s}^{\leq l}) = -\boldsymbol{s}^{\leq l} \boldsymbol{J}_l^{\text{NN}}(\boldsymbol{s}^{\leq l})^\top.$$

Based on RGT (Appendix C), the interaction coefficients $\{J_l\}$ ($J$ in Eq. (2) for each $\boldsymbol{J}_l^{\text{NN}}$) are computed by the following recursive equation in the two-dimensional Ising model (Maris & Kadanoff, 1978): for lattice spacing $a_l = \frac{a_{l-1}}{\sqrt{2}}$, i.e., 45 degree rotated lattice,

$$K_{l-1} = \frac{3}{8} \log\{\cosh(4K_l)\}, \tag{37}$$

where $K_l = \beta J_l$ for $l = L, \ldots, 1$ with $J_L = J$.

Thanks to the site partitioning and the approximation with NN interactions, the conditional sampling probability can be fully decomposed into independent distributions as

$$q^{\mathrm{NN}}(s^l | s^{\leq l-1}) = \prod_{v=1}^{V_l} q^{\mathrm{NN}}(s_v^l | s^{\leq l-1}) \tag{38}$$

(see the Markov blankets shown Figure 1). Sampling from the independent distribution $q^{\mathrm{NN}}(s_v^l | s^{\leq l-1})$ can be easily performed by the heat bath (HB) algorithm: for the Ising models, each site $s_v^l \in \{-1, 1\}$ follows

$$q^{\mathrm{NN}}(s_v^l | s^{\leq l-1}) = \frac{\exp\left(-s_v^l \sum_{v'} (\boldsymbol{J}_l^{\mathrm{NN}})_{v,v'} s_{v'}^{<l-1}\right)}{\exp\left(\sum_{v'} (\boldsymbol{J}_l^{\mathrm{NN}})_{v,v'} s_{v'}^{<l-1}\right) + \exp\left(-\sum_{v'} (\boldsymbol{J}_l^{\mathrm{NN}})_{v,v'} s_{v'}^{<l-1}\right)},$$

and can therefore be sampled by using a probability versus state space table. This is efficient because each site has only $2D$ nearest neighbors. When sampling coarser levels, where the nearest-neighbor interactions are governed by the recursion relation (37), the sampling distribution can be tuned by adjusting $J_l$ to improve the performance.

# E  ALGORITHMIC DETAILS OF RiGCS

In this section, we provide a detailed description of the specialized *sequential training with model transfer* for RiGCS. Furthermore, we also provide pseudocode for both training and sampling.

## E.1  DETAILS OF SEQUENTIAL TRAINING WITH MODEL TRANSFER INITIALIZATION

As mentioned in Section 3.3, we train our RiGCS by minimizing the reverse Kullback-Leibler (KL) divergence (11), which can suffer from long initial random walking steps if the training parameters $\boldsymbol{\theta}$ are not well initialized, e.g., by randomly initialization. This is because a randomly initialized RiGCS, $q_{\boldsymbol{\theta}}(s)$, generates random samples, for which the stochastic gradient of the objective (11) rarely provides useful signal to train the model for a large lattice system. We tackle this problem with a specialized training procedure for RiGCS using *model transfer*, again based on RGT.

We choose $L$ to be an even number, and consider a set of *sequential target* Boltzmann distributions $\{p_{L'}(s^{\leq L'}) \propto e^{-\beta \widetilde{H}_{L'}(s^{\leq L'})}; L' = 0, 2, 4, \ldots, L\}$, where $\{\widetilde{H}_l(s^{\leq l})\}_{l=0}^{L}$ are the approximate Hamiltonians with NN interactions, defined in Eq.(5), and $\widetilde{H}_L(s^{\leq L}) = H(s)$. Let us consider the corresponding set of RiGCS models $\{q_{\boldsymbol{\theta}_0}(s^0), \{q_{\boldsymbol{\theta}_{\leq L'-1}}(s^{\leq L'}); L' = 2, 4, \ldots, L\}\}$ that share the same coarsest lattice size $V_0$. We train each RiGCS model to the sequential targets in the increasing order of $L'$, namely,

At level 0, the 0-layered RiGCS (plain VAN) $q_{\boldsymbol{\theta}_0}(s_0)$ is trained on $p_0(s^0) \propto e^{-\beta \widetilde{H}_0(s^0)}$,

At level 2, the 2-layered RiGCS $q_{\boldsymbol{\theta}_{\leq 1}}(s_{\leq 2}) = p(s^2 | s^{\leq 1}) q_{\boldsymbol{\theta}_1}(s^1 | s^0) q_{\boldsymbol{\theta}_0}(s_0)$

$$\text{is trained on } p_2(s^{\leq 2}) \propto e^{-\beta \widetilde{H}_2(s^{\leq 2})},$$

At level 4, the 4-layered RiGCS $q_{\boldsymbol{\theta}_{\leq 3}}(s_{\leq 4}) = p(s^4 | s^{\leq 3}) \left( \prod_{l=1}^{3} q_{\boldsymbol{\theta}_l}(s^l | s^{\leq l-1}) \right) q_{\boldsymbol{\theta}_0}(s^0)$

$$\text{is trained on } p_4(s^{\leq 4}) \propto e^{-\beta \widetilde{H}_4(s^{\leq 4})},$$

$\vdots$

At level $L'$, the $L'$-layered RiGCS $q_{\boldsymbol{\theta}_{\leq L'-1}}(s_{\leq L'}) = p(s^{L'} | s^{\leq L'-1}) \left( \prod_{l=1}^{L'-1} q_{\boldsymbol{\theta}_l}(s^l | s^{\leq l-1}) \right) q_{\boldsymbol{\theta}_0}(s^0)$

$$\text{is trained on } p_{L'}(s^{\leq L'}) \propto e^{-\beta \widetilde{H}_{L'}(s^{\leq L'})},$$

$\vdots$

At level $L$, the $L$-layered RiGCS $q_{\boldsymbol{\theta}_{\leq L-1}}(\boldsymbol{s}_{\leq L}) = p(\boldsymbol{s}^L|\boldsymbol{s}^{\leq L-1}) \left(\prod_{l=1}^{L-1} q_{\boldsymbol{\theta}_l}(\boldsymbol{s}^l|\boldsymbol{s}^{\leq l-1})\right) q_{\boldsymbol{\theta}_0}(\boldsymbol{s}^0)$

is trained on $p_L(\boldsymbol{s}^{\leq L}) \propto e^{-\beta \widetilde{H}_L(\boldsymbol{s}^{\leq L})}$.

Figure 2 illustrates this procedure in the case of $L = 6$ for the $16 \times 16$ lattice, where the parameters $\{\boldsymbol{\theta}_l\}$ to be trained are explicitly shown.

Now, assume that scale invariance at criticality (SIC) holds *approximately*. Then, the renormalized Hamiltonian should be *well* approximated with NN interactions, i.e., $H_l(\boldsymbol{s}^{\leq l}) \approx \widetilde{H}_l(\boldsymbol{s}^{\leq l})$ (see Eq. (5)). This means that the sequential target distributions $\{p_{L'}(\boldsymbol{s}^{\leq L'}); L' = 0, 2, 4, \ldots, L\}$ have similar marginal distributions on the sites $\boldsymbol{s}^{\leq l}$ (for $l \leq L'$). Therefore, once the parameters $\{\boldsymbol{\theta}_l\}$ of the $(L' - 2)$-layered RiGCS have been trained on the corresponding target $p_{L'-2}(\boldsymbol{s}^{\leq L'})$, they represent suitable initializations for the corresponding components of the $L'$-layered RiGCS to be trained on the next level, i.e., for the target distribution $p_{L'}(\boldsymbol{s}^{\leq L'})$. This justifies the model transfer initializations depicted as the *vertical* red arrows in Figure 2. Furthermore, SIC—stating that the interaction terms in the renormalized Hamiltonians $\{H_l(\boldsymbol{s}^{\leq l})\}$ quickly converge to a fixed point for $l < \widetilde{L}$ with some $\widetilde{L} < L$—implies that the renormalized Hamiltonians for different scales, e.g., $H_{l-2}(\boldsymbol{s}^{\leq l-2})$ and $H_l(\boldsymbol{s}^{\leq l})$, should consist of similar sets of interaction terms. Therefore, thanks to our choice of using the same architecture for all conditional models over different levels, we can also apply model transfer initializations from $\boldsymbol{\theta}_{l-2}$ to $\boldsymbol{\theta}_l$, as depicted as the *diagonal* red arrows in Figure 2.

In summary, our sequential training with model transfer follows the following procedure:

1. Train the (unconditional) generative model $q_{\boldsymbol{\theta}_0}(\boldsymbol{s}_0)$ to approximate $\widetilde{p}(\boldsymbol{s}^0) \propto e^{-\beta \widetilde{H}_0(\boldsymbol{s}^0)}$ with $\widetilde{\boldsymbol{J}}_0^{\mathrm{NN}}$. Set $\widetilde{\boldsymbol{\theta}}_0 \leftarrow \boldsymbol{\theta}_0$.

2. Refine $\boldsymbol{\theta}_{\leq 1}$ from its initial value $\widetilde{\boldsymbol{\theta}}_{\leq 1} = (\widetilde{\boldsymbol{\theta}}_1, \widetilde{\boldsymbol{\theta}}_0)$, where $\widetilde{\boldsymbol{\theta}}_1$ is set randomly, by training $q_{\boldsymbol{\theta}_{\leq 1}}(\boldsymbol{s}_{\leq 2}) = p(\boldsymbol{s}^2|\boldsymbol{s}^{\leq 1})q_{\boldsymbol{\theta}_1}(\boldsymbol{s}^1|\boldsymbol{s}^0)q_{\boldsymbol{\theta}_0}(\boldsymbol{s}^0)$ to approximate $\widetilde{p}(\boldsymbol{s}^{\leq 2}) \propto e^{-\beta \widetilde{H}_2(\boldsymbol{s}^{\leq 2})}$. Set $\widetilde{\boldsymbol{\theta}}_{\leq 1} \leftarrow \boldsymbol{\theta}_{\leq 1}$.

3. Refine $\boldsymbol{\theta}_{\leq 3}$ from its initial value $\widetilde{\boldsymbol{\theta}}_{\leq 3} = (\widetilde{\boldsymbol{\theta}}_1, \widetilde{\boldsymbol{\theta}}_2, \widetilde{\boldsymbol{\theta}}_1, \widetilde{\boldsymbol{\theta}}_0)$, where $\widetilde{\boldsymbol{\theta}}_2$ is set randomly, by training $q_{\boldsymbol{\theta}_{\leq 3}}(\boldsymbol{s}_{\leq 4}) = p(\boldsymbol{s}^4|\boldsymbol{s}^{\leq 3}) \left(\prod_{l'=1}^3 q_{\boldsymbol{\theta}_{l'}}(\boldsymbol{s}^{l'}|\boldsymbol{s}^{l'-1})\right) q_{\boldsymbol{\theta}_0}(\boldsymbol{s}^0)$ to approximate $\widetilde{p}(\boldsymbol{s}^{\leq 4}) \propto e^{-\beta \widetilde{H}_4(\boldsymbol{s}^{\leq 4})}$. Set $\widetilde{\boldsymbol{\theta}}_{\leq 3} \leftarrow \boldsymbol{\theta}_{\leq 3}$.

4. For $L' = 6, 8, \ldots, L$, refine $\boldsymbol{\theta}_{\leq L'-1}$ from its initial value $\widetilde{\boldsymbol{\theta}}_{\leq L'-1} = (\widetilde{\boldsymbol{\theta}}_{L'-3}, \widetilde{\boldsymbol{\theta}}_{L'-4}, \widetilde{\boldsymbol{\theta}}_{\leq L'-3})$ by training $q_{\boldsymbol{\theta}_{\leq L'-1}}(\boldsymbol{s}_{\leq L'}) = p(\boldsymbol{s}^{L'}|\boldsymbol{s}^{\leq L'-1}) \left(\prod_{l=1}^{L'-1} q_{\boldsymbol{\theta}_l}(\boldsymbol{s}^l|\boldsymbol{s}^{\leq l-1})\right) q_{\boldsymbol{\theta}_0}(\boldsymbol{s}^0)$ to approximate $\widetilde{p}(\boldsymbol{s}^{\leq L'}) \propto e^{-\beta \widetilde{H}_{L'}(\boldsymbol{s}^{\leq L'})}$. Set $\widetilde{\boldsymbol{\theta}}_{\leq L'-1} \leftarrow \boldsymbol{\theta}_{\leq L'-1}$.

Note that all parameters except $\boldsymbol{\theta}_0, \boldsymbol{\theta}_1, \boldsymbol{\theta}_2$—which are trained on the three smallest lattice sizes with random initializations—can be initialized to the parameters trained on easier problems (smaller lattice), which significantly accelerates the training process, as shown in Figure 3 (right). For large $L$ and $l \ll L$, the approximate renormalized Hamiltonian $\widetilde{H}_l(\boldsymbol{s}^{\leq l})$ with NN interactions might be significantly different from the true renormalized Hamiltonian $H_l(\boldsymbol{s}^{\leq l})$ that may have long-range and higher-order interaction terms. Crucially, our training procedure helps reducing this gap step by step by fine-tuning the generative models with receptive fields extending beyond nearest neighbors.

### E.2 PSEUDOCODE FOR RiGCS

The pseudocodes provided in Algorithm 1 and Algorithm 2 describe the practical steps for training RiGCS and for sampling from a trained RiGCS, respectively.

## F AUTOREGRESSIVE NEURAL NETWORKS

Autoregressive neural networks are a class of generative models, where the elements of random configurations are ordered, and each element is sampled, depending only on the *previous* elements.

---

**Algorithm 1** RiGCS training

---

1: **Input:**
- Coarsest lattice size $N_0$
- PixelCNN $q_{\theta_0}$
- numbers of levels $L$
- Conditional networks $\{q_{\theta_l}(s^l|s^{\leq l-1})\}$
- HB algorithm $p(s^l|s^{l-1})$

2: **Output:**
- **Trained RiGCS** (PixelCNN-based generative model): list of conditional models for sampling $s^L \in \mathcal{S}^D$ with $D = 2^L N_0 \times 2^L N_0$

3: Train the PixelCNN $q_{\theta_0}$ on $N_0 \times N_0$ target lattices.
4: Add to the RiGCS's list of models the conditional VAN $q_{\theta_1}(s^1|s^{\leq 0})$ (randomly initialized) and the HB $p(s^2|s^{\leq 1})$.
5: Train the RiGCS on $2N_0 \times 2N_0$ lattices.
6: Replace the HB $p(s^2|s^{\leq 1})$ with the conditional VAN $q_{\theta_2}(s^2|s^{\leq 1})$ randomly initialized.
7: Add to the RiGCS's list of models the conditional VAN $q_{\theta_3}(s^3|s^{\leq 2})$ initialized with the weights of the trained model $q_{\widetilde{\theta}_1}$, and the HB $p(s^4|s^{\leq 3})$.
8: Train the RiGCS on $4N_0 \times 4N_0$ lattices.
9: **for** $l = 5, l < L, l = l + 2$ **do**
10:     Replace the HB $p(s^{l-1}|s^{\leq l-2})$ with the conditional VAN $q_{\theta_{l-1}}(s^{l-1}|s^{\leq l-2})$ initialized with the trained model $q_{\widetilde{\theta}_{l-3}}$ weights.
11:     Add to the RiGCS's list of models the conditional VAN $q_{\theta_l}(s^l|s^{\leq l-1})$, initialized with the trained model $q_{\widetilde{\theta}_{l-2}}$ weights, and HB $p(s^{l+1}|s^{\leq l})$.
12:     Train the RiGCS on lattices of size $2^{l+1}N_0 \times 2^{l+1}N_0$.
13: **end for**
14: **return** List of (trained) conditional models $\{q_{\theta_l}(s^l|s^{\leq l-1})\}_{l=1}^{L-1}$

---

These models are widely used in time series forecasting (Triebe et al., 2019), natural language processing (van den Oord et al., 2016a), large language models (Brown et al., 2020), and generative modeling (van den Oord et al., 2016b) as they explicitly capture the dependencies between elements in a sequence.

In the last decades, autoregressive neural networks have been extensively deployed in different scientific domains including statistical physics (Wu et al., 2019; Nicoli et al., 2020; Wang et al., 2022; Biazzo, 2023; Biazzo et al., 2024), quantum chemistry (Gebauer et al., 2019; Joshi et al., 2021; Gebauer et al., 2022), learning wave functions of many body systems (Hibat-Allah et al., 2020), tensor newtorks (Chen et al., 2023) and quantum computing (Liu et al., 2021).

Relying on the factorizability of arbitrary distributions as

$$p(s) = \left( \prod_{v=2}^{V} p(s_v \mid s_{v-1}, \ldots, s_1) \right) p(s_1), \tag{39}$$

autoregressive models approximate each factor in the right-hand side with neural network models $q_\theta$ with the parameters $\theta$ to be optimized so that $q_\theta(s) \approx p(s)$. The ancestral sampling in the order of $s_1, \ldots, s_V$ allows sampling and density evaluation at the same time. State-of-the-art architectures often use convolutional neural networks, leveraging masked filters to ensure that the conditional dependencies are restricted to previous elements in the sequence (van den Oord et al., 2016b; Salimans et al., 2017).

---

**Algorithm 2** RiGCS sampling

---

1: **Input:**
- Coarsest lattice size $N_0$
- PixelCNN $q_{\theta_0}$
- List of conditional models $\{q_{\theta_l}(s^l|s^{\leq l-1})\}_{l=1}^{L-1}$
- Heatbath algorithm for sampling the finest level $L$: $p(s^L|s^{L-1})$.

2: **Output:**
- **Samples**: $s^L \in \mathcal{S}^D$ with $D = 2^L N_0 \times 2^L N_0$
- **Log-prob**: Exact sampling probability $\ln q_\theta(s^L)$.

 

3: Sample $s^0 \sim q_{\theta_0}$ and compute $\ln q_{\theta_0}(s_0)$.
4: **for** $l = 1, l < L, l = l + 2$ **do**
5:     Embed the sample $s^{l-1}$ into a $2N_{l-1} \times 2N_{l-1}$ with zeros in the lattice sites of the levels $l, l+1$.
6:     Sample $s_l \sim q_{\theta_l}(s^l|s^{\leq l-1})$ and compute $\ln q_{\theta_l}(s_l)$.
7:     **if** $l + 1 \neq L$ **then**
8:        Sample $s_{l+1} \sim q_{\theta_{l+1}}(s^l|s^{\leq l})$ and compute $\ln q_{\theta_{l+1}}(s^{l+1})$.
9:     **else**
10:       Sample $s^L \sim p(s^L|s^{L-1})$ with HB and compute $\ln q_\theta(s^L)$.
11:     **end if**
12: **end for**
13: **return** $s_L, \ln q_\theta(s^L)$

---

# G   IMPLEMENTATION DETAILS FOR VAN, HAN AND RIGCS

## G.1   VAN

Two possible architectures often used for implementing VAN-like networks (Wu et al., 2019) are Masked Autoencoder for Distribution Estimation (MADE) (Germain et al., 2015) and Pixel-CNN (van den Oord et al., 2016c;b), which rely on fully-connected and convolutional layers, respectively. In order to ensure the autoregressive properties, the weights of these architectures are masked (see Figure 1 in van den Oord et al. (2016b)) so that the $v$-th entry of the output $\widehat{s}_v = g_\theta(s)$ of the network $g_\theta$ depends only on the previous values $s_{<v} = (s_{v-1}, \ldots, s_1)$, i.e.,

$$\widehat{s}_v = g_\theta(s_{<v}).$$

With the sigmoid function used in the last layer of the network $g_\theta(s_{<v})$, the output is bounded as $\widehat{s}_v \in (0, 1)$, which expresses the probability that the entry has the positive spin $s_v = +1$. Namely, the corresponding entry is drawn from the following Bernoulli distribution:

$$q_\theta(s_v|s_{<v}) = \widehat{s}_v \delta_{s_v,+1} + (1 - \widehat{s}_v)\delta_{s_v,-1}. \tag{40}$$

In our experiments, we used the PixelCNN-based VAN—implemented by Wu et al. (2019)—which leverages masked convolutional kernels of the size of $K \times K$ for odd $K$. Let $M \in \{0, 1\}^{K \times K}$ be the mask for the kernel. Noting that $(k, k') = ((K - 1)/2 + 1, (K - 1)/2 + 1)$ corresponds to the center of the kernel, setting the mask such that

$$M_{k,k'} = \begin{cases} 1 & \text{if } [k < (K - 1)/2 + 1] \vee [[k = (K - 1)/2 + 1] \wedge [k' < (K - 1)/2 + 1]], \\ 0 & \text{otherwise} \end{cases}$$

ensures the autoregressive properties. When more than two layers are used, PixelCNN uses residual connections (He et al., 2016) (i.e., the original input to the layer is summed to the output) for each layer, except for the first and the last layers. Within the residual connections before each masked convolutional layer, as well as at the end of the network before the sigmoid function, a standard convolutional layer with the kernel size of $1 \times 1$ is added.

The PixelCNN, used in our experiments as the plain VAN, has 6 masked convolutional layers with 32 channels and the kernel size of $K = 13$.

## G.2 HAN

The HAN (Białas et al., 2022) model leverages recursive domain decomposition (Cè et al., 2016) in order to sample in parallel different regions of the lattice configurations. The crucial aspect of the domain decomposition is that the domains must be connected through a common boundary, and, once it is given, each domain can be sampled independently by using the same model. In the HAN implementation, a boundary $\mathcal{B}_0$ that divides the lattice into four domains is first sampled by using a standard MADE architecture. Then, each domain is further split into four by sampling in parallel four boundaries $\{\mathcal{B}_i\}$ using another MADE model conditioned on the boundary $\mathcal{B}_0$. This procedure is repeated until the remaining entries have all the neighbors fixed and therefore can be sampled by HB.

In our experiments, where HAN is evaluated as a baseline, we used the code and the hyperparameters provided by the authors of Białas et al. (2022).

## G.3 RIGCS

In our implementation of RiGCS, the (unconditional) generative model $q_{\theta_0}(s_0)$ at the coarsest level $l = 0$, consists of a VAN with 3 masked convolutional layers with 12 kernels of the size of $K = 13$.

For the conditional models $\{q_{\theta_l}(s^l|s^{\leq l-1})\}_{l=1}^{L-1}$, we define one "block" as *two* consecutive levels, which correspond to upsampling from lattice size $N_l \times N_l$ to $N_{l+2} \times N_{l+2} = 2N_l \times 2N_l$ for $l \in \{0, 2, 4, \cdots, L - 2\}$. Each block takes as input a coarser configuration $s^{\leq l}$ of size $N_l \times N_l$ and embeds it into a $2N_l \times 2N_l$ configuration where all unsampled entries for the levels $l + 1$ and $l + 2$ are set to 0. Then, the spins $s^{l+1}$ and $s^{l+2}$ are sampled sequentially according to the output of a standard CNN that takes as input the embedded configuration. Each conditional CNN (conditional VAN) of RiGCS has one hidden layer with 12 kernels of the sizes of $K = 5$ and $K = 3$, respectively, for the hidden and output layers, making its receptive field size $7 \times 7$. Within each block, we use the same CNN architecture for both levels. The same architecture applies to all blocks except the last. In the last level, i.e., $l = L$, RiGCS performs the HB sampling because the target Hamiltonian has only nearest-neighbor interactions by assumption.

Similarly to the VAN (see Appendix G.1), the conditional VAN uses a sigmoid activation in the final layer, such that the output is bounded as $\widehat{s}_v \in (0, 1)$. With this output, the spin is drawn from the Bernoulli distribution:

$$q_\theta(s_v^l|s_{<v}^l, s^{<l}) = \widehat{s}_v^l \delta_{s_v^l, +1} + (1 - \widehat{s}_v^l)\delta_{s_v^l, -1}.$$

During the training procedure described in Section 3.3, the weights in each block are initialized with those of the conditional VANs in the coarser block.

## G.4 TRAINING

All the generative neural samplers (RiGCS, VAN, HAN) used in our experiments are trained by minimizing the reverse Kullback-Leibler (KL) divergence

$$\min_{\boldsymbol{\theta}} \mathrm{KL}(q_{\boldsymbol{\theta}}(\boldsymbol{s})\|p(\boldsymbol{s})) \tag{41}$$

with the gradient estimator

$$\nabla_\theta \mathrm{KL}(q_{\boldsymbol{\theta}}(\boldsymbol{s})\|p(\boldsymbol{s})) = \mathbb{E}_{\boldsymbol{s}\sim q_\theta}\big[\big(\beta H(s) + \log q_{\boldsymbol{\theta}}(\boldsymbol{s})\big)\nabla_\theta \log q_\theta(\boldsymbol{s})\big].$$

In order to make the variance of the estimator more stable, we leverage a control variates method (Mnih & Gregor, 2014) as suggested in Wu et al. (2019). We use the ADAM optimizer (Kingma & Ba, 2015) with learning rate 0.001 and standard $\beta$s for training all models.

We trained VANs for 50000 gradient updates (steps) with batch size 100, and HANs for 100000 gradient updates with batch size 1000. For RiGCS, training is performed for a total of 3000 steps for each sequential (upscaled) target lattice. When training on a target lattice $N_L = N$, the pretraining phase involves training at coarser levels as follows: 2000 steps for level $L - 2$, 1500 steps for level $L - 4$, and 1000 steps for all previous levels, except for the coarsest one which is always trained for 500 steps.

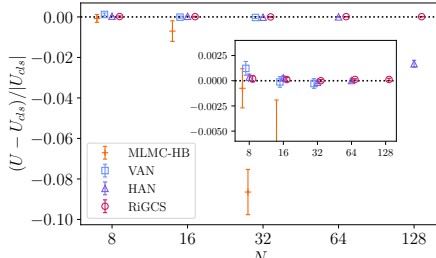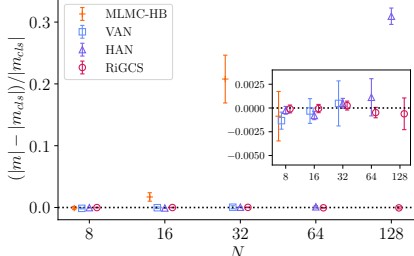

Figure 7: Relative estimation error for the internal energy (left) and absolute magnetization (right). The vanilla VAN cannot be trained for $N \geq 64$ in reasonable time. Note that, unlike in Figure 3 (left), we used the MC estimators by the Cluster-AIS $U_{\mathrm{cls}}$ and $m_{\mathrm{cls}}$ as the reference values, as the ground truth cannot be computed analytically.

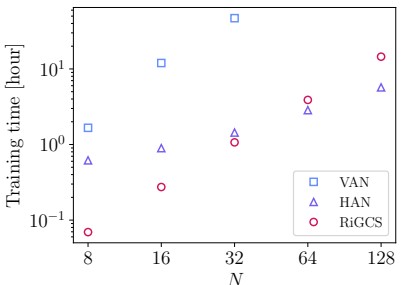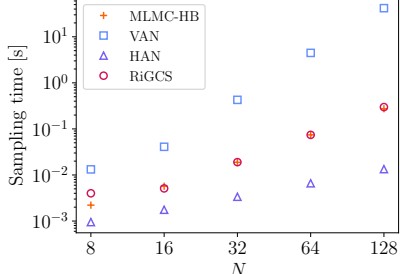

Figure 8: Total training time (left) and sampling time (right) for different lattice sizes.

### G.5 SAMPLING

We computed the MC estimates with one million samples for the Cluster(-AIS), HAN and RiGCS. For VAN and MLMC-HB we used 100k samples. For the largest lattice considered in this work, i.e., $128 \times 128$, Cluster-AIS sampled only 500k configurations due to its high computational costs. Errors are estimated using an automatic differentiation method introduced in Ramos (2019) and implemented by Joswig et al. (2023).

## H ADDITIONAL NUMERICAL RESULTS

### H.1 NON-THERMODYNAMIC OBSERVABLE ESTIMATION

Figure 7 shows additional numerical results of estimating the internal energy and the magnetization, where the estimators by the cluster method are used as the reference values. Consistently with the EMDM and ESS shown in Figure 4, our RiGCS provides unbiased estimates with lowest variances compared to other generative neural samplers.

### H.2 COMPUTATION TIME

Figure 8 shows empirical training (left) and sampling (right) time. For all models (RiGCS and the baselines), we used a single NVIDIA A100 GPU with 80 GB of memory.

### H.3 ABLATION STUDY: TRANSFORMER ARCHITECTURE FOR CONDITIONAL VAN

In order to investigate the impact of different neural network architectures for the conditional VAN used in RiGCS, we compared the performances between CNN- and transformer-based neural networks. For the latter, we replaced the CNN of our proposed implementation with a transformer (encoder only) with 5 multi-head attention block (8 heads) and 1024 neurons in the feed-forward

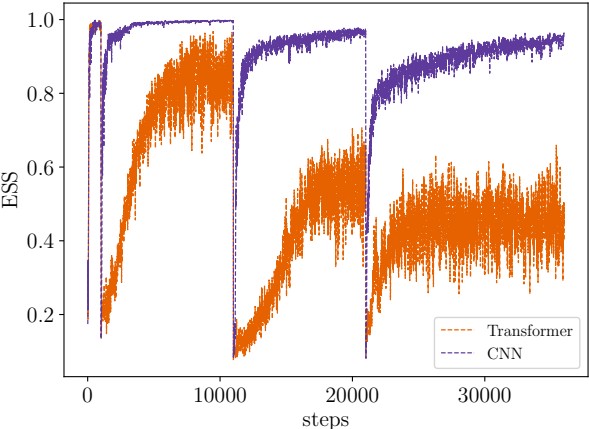

Figure 9: Comparison between a CNN-based RiGCS (purple) and a transformer-based RiGCS (orange). Both CNN and transformer refer to the type of neural network architectures used for the conditional models in RiGCS. Both models use our proposed warm-start technique for training for the target (finest) lattice size of $16 \times 16$.

networks. However, for the unconditional VAN at the coarsest level $l = 0$, we kept using the PixelCNN. Figure 9 displays the results of sequential training with model transfer up to lattices of size $16 \times 16$. Note that, unlike in Figure 3 (right), the plot shows all training procedure including the pretraining phase. Specifically, the initial phase (steps: 1 to 1001) corresponds to training of the unconditional VAN at $l = 0$, the second (steps: 1002 to 11002) and the third (steps: 11003 to 21003) phases involve training of the conditional VANs at $l = 1, 2$, respectively, and the last phase (steps: 21004 to the end of training) corresponds to training of the full RiGCS for the target lattice of size $16 \times 16$.

We observe that, except the initial phase, where both RiGCS models use the same unconditional VAN with PixelCNN, RiGCS with CNN (purple) clearly outperforms RiGCS with Transformer (orange). This result is consistent with recent work (Abbott et al., 2023; Liu et al., 2024), where transformer-based architectures do not appear to be well-suited for lattice simulations of physical systems, despite their great success across various domains. The reasons for this remain to be theoretically understood.

### H.4 PRELIMINARY EXPERIMENT FOR LATTICE QUANTUM FIELD THEORIES

To further validate our approach, we tested RiGCS on another popular benchmark in computational physics: the $\phi^4$ scalar field theory. This field theory, when discretized on a lattice, serves as an important benchmark for validating sampling algorithms. In fact, it has been extensively studied in the literature (Albergo et al., 2019; Nicoli et al., 2021) for benchmarking the performance of generative models (e.g., normalizing flows) with continuous degrees of freedom. The $\phi^4$ field theory belongs to the same universality class as the Ising model (i.e., as discussed in Appendix C, they share the same critical exponents). Despite being recognized as a toy model in the 2-dimensional lattice, the $\phi^4$ theory is of high-relevance to physicists for the following reasons. In the *standard model of particle physics*, the field of the *Higgs boson* possesses the same type of interaction as described by the $\phi^4$ lattice field theory. Moreover, the $\phi^4$ scalar field theory exhibits spontaneous symmetry breaking of the $\mathbb{Z}_2$ invariance, which is a property central to the Higgs mechanism as well.

Configurations of the scalar field theory are sampled from the Boltzmann distribution

$$p(\phi) = \frac{e^{-S[\phi]}}{Z},$$

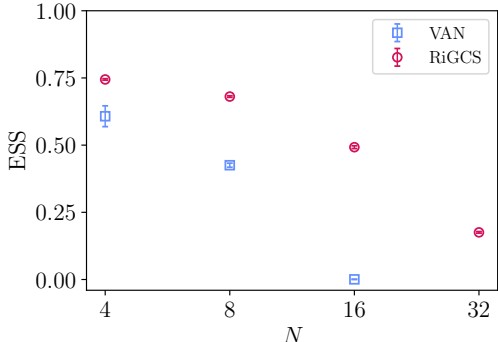

Figure 10: Effective Sample Size (ESS) for different lattice sizes for the $\phi^4$ theory, after 5k steps of training for $N = \{4, 8, 16\}$ and 10k for $N = 32$. Blue squares and magenta circles refer to the plain VAN (baseline) and RiGCS (ours), respectively. The plain VAN can be trained in reasonable time only up to $16 \times 16$ lattices, and RiGCS consistently achieves a higher ESS than VAN.

where $Z$ is the unknown partition function, and $S[\phi]$ is the *action*—a functional describing the physics of the system in the Lagrangian formalism. For the $\phi^4$ theory, the action reads

$$S[\phi] = \sum_v \left[ -2\kappa \sum_{\mu=1}^{D} \phi_v \phi_{v+\mu} + (1-2\lambda)\phi_v^2 + \lambda \phi_v^4 \right], \tag{42}$$

where $\kappa$ is the hopping parameter controlling the nearest-neighbor interactions, and $\lambda$ is the bare coupling parameter determining the strength of the quartic self-interactions. The index $\mu$ labels the basis vectors in the $D$-dimensional space.

In the limit of $\lambda \to \infty$, the self-interaction terms enforce $\phi_v \to \pm 1$, effectively reducing the field to discrete spin values $s_v = \pm 1$ (Vierhaus, 2010; Wolff, 2009). The resulting effective action in this limit corresponds to the ferromagnetic Ising Hamiltonian

$$S_E[\boldsymbol{s}] = H(\boldsymbol{s}) = -\beta \sum_{\langle v,v' \rangle} s_v s_{v'}, \quad \text{for} \quad \beta = 2\kappa. \tag{43}$$

In the limit of $\lambda \to 0$, the quartic interaction vanishes, and the theory reduces to a free (Gaussian) field theory. We refer to Vierhaus (2010); Wolff (2009) for further details on the connection between the Ising model and the $\phi^4$ theory.

The architecture of RiGCS used in our preliminary experiment for simulating the $\phi^4$ theory closely follows that for the Ising model used in the main paper, with a few key differences outlined below. As in the Ising case, we use a PixelCNN (plain VAN) at the coarsest level, and conditional CNNs at the intermediate levels. However, unlike for the Ising model, we also use the conditional CNN, instead of the HB sampler, at the finest level. For sampling continuous variables, we use Gaussian Mixture Models (GMMs), following the approach by Faraz et al. (2025). Specifically, at level $l$, GMM represents the conditional probability density for a new site $\phi_v^l$ in the configuration as

$$q_{\boldsymbol{\theta}_l}(\phi_v^l | \boldsymbol{\phi}_{<v}^l, \boldsymbol{\phi}^{<l}) = \sum_{h=1}^{H} \pi_{h,v}^l \mathcal{N}(\phi_v^l | \mu_{h,v}^l, \sigma_{h,v}^l).$$

Here, $\pi_{h,v}^l = \pi_h(\boldsymbol{\phi}_{<v}^l, \boldsymbol{\phi}_{<l}; \boldsymbol{\theta}_l)$, $\mu_{h,v}^l = \mu_h(\boldsymbol{\phi}_{<v}^l, \boldsymbol{\phi}_{<l}; \boldsymbol{\theta}_l)$, and $\sigma_{h,v}^l = \sigma_h(\boldsymbol{\phi}_{<v}^l, \boldsymbol{\phi}_{<l}; \boldsymbol{\theta}_l)$ are neural networks, which are parameterized with $\boldsymbol{\theta}_l$, and represent the mixing coefficient, the mean, and the standard deviation of the $h$-th Gaussian component, respectively. We set $H = 3$.

We perform sequential training similar to the Ising model case with a minor modification: since we use conditional CNN at the finest level, all HB blocks in Figure 2 are replaced with conditional CNNs parameterized by $\boldsymbol{\theta}_{L'}$. More specifically, the HB block at level 2 is replaced with $\boldsymbol{\theta}_2$, which

is initialized randomly. The trained $\boldsymbol{\theta}_2$ is transferred to $\boldsymbol{\theta}_2$ and $\boldsymbol{\theta}_4$ (substituted for HB) at Level 4. We continue this model transfer to Level 6 and higher.

We conducted preliminary experiments for comparing the performance of VAN and RiGCS in sampling configurations for the $\phi^4$ theory at the critical point. We evaluated the Effective Sample Size (ESS) for lattices up to $32 \times 32$, and plotted the results in Figure 10. We observe that RiGCS again consistently outperforms the plain VAN (Wu et al., 2019), always converging to a higher ESS. Furthermore, for lattices larger than $32 \times 32$, the plain VAN could not be trained due to excessive computational costs, while RiGCS can be trained with reasonable computational costs. This further highlights the superior performance and efficiency of RiGCS. We note that the VAN architecture— which is particularly suited for sampling discrete variables—is not necessarily an optimal choice for sampling continuous variables. Exploring RiGCS with more suitable base generative samplers, e.g., normalizing flows, for continuous systems is a promising direction future research.

