# OpenReview forum: "Multilevel Generative Samplers for Investigating Critical Phenomena"
_ICLR.cc/2025/Conference — ICLR 2025 Poster_

### Official Review · Reviewer_pv8Q · 2024-10-26

**Soundness:** 3
**Presentation:** 2
**Contribution:** 2
**Rating:** 6
**Confidence:** 2

**Summary:**

This paper introduces the Renormalization-informed Generative Critical Sampler (RiGCS), a novel generative sampling algorithm designed for near-critical systems where scale-invariant criticality (SIC) can hinder the efficiency of standard Monte Carlo simulations. The proposed RiGCS builds on Multilevel Monte Carlo algorithms with Heat Bath sampling and incorporates generative models to address the shortcomings of HB under slight SIC violations.

**Strengths:**

1. The proposed method integrates renormalization group theory with generative models in a novel attempt to address critical slowing down in Monte Carlo simulations.
2. Employing generative models to enhance MLMC-HB sampling seems to be a promising approach with potential applications in statistical physics and lattice systems.

**Weaknesses:**

1. The core contributions (lines 90–99) are not sufficiently clear. See Questions 1-3.
2. The mathematical formulations are not clear. For example, Equation (9) is not well defined and difficult to interpret. The algorithm design between lines 355–369 is unclear, and the lack of theoretical guarantees on convergence or sample complexity weakens the technical rigor.
3. The experiments are limited in scope. To demonstrate generalizability, the authors should evaluate RiGCS on more complex benchmarks beyond the 2D Ising model.
4. The appendix introduces unrelated terminologies (e.g., Swendsen-Wang algorithm, Wolff algorithm) without meaningful connections to the core work or further experimental analysis. Moreover, the lack of theoretical support or detailed experimental setups in the appendix limits its usefulness.
5. The manuscript cites more than 100 references, many of which seem only loosely related to the topic.

**Questions:**

1. Why is addressing SIC important in this context? While the authors claim that SIC reduces sampling efficiency, they do not clearly explain how RiGCS specifically addresses this issue. A more detailed justification is required.
2. Although the warm start is listed as a key contribution, the implementation details are unclear. How is the warm start applied across different resolution levels, and why is it considered a significant innovation?
3. If the authors only employ standard metrics (e.g., effective sample size, free energy bias), it is unclear why performance evaluation is listed as a major contribution. Were any new metrics introduced? If not, the framing of performance evaluation as a contribution should be reconsidered.

---

> ### Author Response · Authors · 2024-11-22
> **Reply to Reviewer pv8Q 1/3**
>
> We appreciate the reviewer’s comments, which highlighted readability issues in the submitted version. We are committed to addressing these and will make every effort to improve the manuscript.
> We first refer the reviewer to the general reply, which includes an important correction to the experimental results in Fig.2 (left).  Below we address the specific questions and concerns. Please note that we have not uploaded a revision yet, but we plan on doing it before the discussion phase ends.
>
> > Weakness 1: The core contributions (lines 90–99) are not sufficiently clear. See Questions 1-3.
>
> > Q 1: Why is addressing SIC important in this context? While the authors claim that SIC reduces sampling efficiency, they do not clearly explain how RiGCS specifically addresses this issue. A more detailed justification is required.
>
> - **Why is addressing SIC important:** The reason is nicely described by Reviewer cr5x in the summary part.  Taking their words, we will reinforce our explanation in Section 1 by adding the following: “At the criticality, physical systems typically exhibit self-similarity with respect to the change of scale, i.e., the physics of the coarse-grained system is similar to that of the fine-grained one. This requires us to deal with arbitrarily long-range correlations for which standard MCMC samplers based on local moves undergo critical slowing down, meaning that the auto-correlation time becomes arbitrarily large with system size.”
>
> - **How RiGCS specifically addresses this issue:**  In general, any distribution can be decomposed as Eq.(7).  Although the left-hand side $p(s)$ exhibits long-range correlations, each of the factors $\{p(s^l|s^{\leq l-1}\)}$ and $p(s^0)$ in the right-hand side exhibit only short-range correlations (with respect to the corresponding resolution scale), according to the renormalization group theory.  RiGCS approximates not directly $p(s)$ with long-range correlations but each factor in the right-hand side with short-range correlations, using generative models with small receptive fields.  We can intuitively say that *RiGCS addresses the issue, i.e., long-range correlations induced by SIC, by capturing the long-range correlations as short-range correlations in the coarse level marginal distributions.*
>
> > Q2: Although the warm start is listed as a key contribution, the implementation details are unclear. How is the warm start applied across different resolution levels, and why is it considered a significant innovation?
>
> - **Clarity:** We agree with the reviewer that the implementation details of the training procedure were unclear.  We will improve the explanation and add an illustration of the workflow as well as a detailed pseudocode.
> - **Innovation:** We argue that our warm start (sequential) training with model transfer is a significant innovation as a *novel physics-informed training procedure*.  First of all, model initialization is crucial when training a model by minimizing the reverse KL divergence because samples from badly initialized models do not provide informative gradient signals for training.  The situation gets exponentially worse for larger lattice systems.  Our sequential training with model transfer leverages SIC, i.e., the self-similarity across scales, and starts from training models for small lattice sizes, e.g., $2\times 2$ and $4\times 4$. Then, to train a larger model, e.g., $8 \times 8$, each conditional model in Eq.(7) has a similar conditional model already trained for smaller lattice sizes, which can be used for initialization. This idea is novel to the best of our knowledge and effective according to our experiment shown in Fig. 2 right. We will improve the description of this training procedure (see below).
>
> > Q3: If the authors only employ standard metrics (e.g., effective sample size, free energy bias), it is unclear why performance evaluation is listed as a major contribution. Were any new metrics introduced? If not, the framing of performance evaluation as a contribution should be reconsidered.
>
> We moved the corresponding sentence outside the list of contributions in the revised manuscript.

---

> > ### Author Response · Authors · 2024-11-22
> > **Reply to Reviewer pv8Q 2/3**
> >
> > > W2: The mathematical formulations are not clear. For example, Equation (9) is not well defined and difficult to interpret. The algorithm design between lines 355–369 is unclear, and the lack of theoretical guarantees on convergence or sample complexity weakens the technical rigor.
> >
> > - **Eq.(9)**: this equation is indeed not well-defined because it only expresses a general form without specifying each term in the expansion.  We supposed that the interpretation was clear with the superscript for the coefficients, and the orders of the corresponding terms, i.e., the second-order nearest neighbor interaction term, the second-order long-range interaction term, and the third-order interaction terms with the dots implicitly expressing higher-order interaction terms.  This equation should be understood like Taylor decompositions with infinitely many terms. We refer the reviewer to Appendix C where the details of the renormalization group are introduced. We will add a sentence to the revised manuscript to avoid any confusion.
> >
> > - **algorithm design between lines 355–369:** We agree that the description of the training procedure 355-369 is indeed hard to read, although mathematically precise.  In the revised manuscript, we will add an illustration to complement this description  and a pseudocode of the training procedure, to enhance a more intuitive understanding of the training process.
> >
> > - **technical rigor:** We respectfully disagree with the reviewer about the weak technical rigor.  The basic building block is ARNN, for which the training is known to converge, and the whole framework of generative MC simulation is guaranteed to be unbiased [1], thus providing a reliable confidence interval for the observable estimation as long as the mode-dropping does not occur.  For this reason, we also analyzed the effective mode-dropping measure. In the manuscript, we show that RiGCS did not suffer from any mode-dropping (see Fig. 3 left), and we report its unbiased estimates for free energy, internal energy, and magnetization in Figs. 2 (left) and 5 (left and right) respectively.  We would appreciate it if the referee could specify which theory is required to maintain technical rigor.
> >
> > > W3: The experiments are limited in scope. To demonstrate generalizability, the authors should evaluate RiGCS on more complex benchmarks beyond the 2D Ising model.
> >
> > We are implementing new theories to benchmark our method.  We plan to update the revision with new results before the discussion phase ends (see also general rebuttal).

---

> ### Author Response · Authors · 2024-11-22
> **Reply to Reviewer pv8Q 3/3**
>
> > W4: The appendix introduces unrelated terminologies (e.g., Swendsen-Wang algorithm, Wolff algorithm) without meaningful connections to the core work or further experimental analysis. Moreover, the lack of theoretical support or detailed experimental setups in the appendix limits its usefulness.
>
> - **Appendix on Cluster Algorithms**: Appendix B.2 is meant to be a self-contained introduction to two popular Cluster algorithms, one of which is used in our experiment as the “cluster algorithm”.  As the referee pointed out, the connection is unclear because we did not mention that the Wolff algorithm, effectively a cluster algorithm, is chosen as the cluster method in the main text.  We will clarify these connections in the revised manuscript.
>
> - **Lack of theoretical support**:  We do not see what kind of *theoretical support* is necessary to support our method.  As mentioned above, the training of RiGCS is guaranteed to converge because the training of ARNN is known to converge. The generative MC simulation with importance re-weighting is guaranteed to be unbiased [1] unless mode-dropping occurs.  Our experiments empirically verify that RiGCS does not show mode-dropping.  We request the reviewer to please concretely specify what kind of “lack of theoretical support” limits the usefulness of our method.
>
> - **Detailed experimental setups**: We detailed the experimental setup for our numerical analysis in Appendix F. In the revised version of the manuscript we will also add pseudocodes for RiGCS.  We would appreciate it if the reviewer could specify what relevant details are missing and we will be happy to add them in the revised manuscript.
>
> > W5: The manuscript cites more than 100 references, many of which seem only loosely related to the topic.
>
> We agree that we cited many papers, including weakly related works.  However, this is because we decided to have an “Extended related work” section in the Appendix, where the general influence of renormalization group theory (RGT) on machine learning is discussed.  We argue that this part is useful for readers who are not necessarily familiar with the topic but may be interested in RGT and would consider incorporating it for their future machine learning research. By quoting the words of reviewer cr5x, we stress that our goal was to provide readers with *“​​[..] an impressive bibliography on the subject which is indeed quite vast and old. A synthetic and pedagogical section in the supplementary material, which should be useful to non-physicists, is devoted to the RG theory and should make the whole paper readable for non-physicists”*.  However, we found that we did not clearly describe some of the references, and therefore, we revised the appendix so that the (either strong or weak) relations between RiGCS and other works become clear.

---

> > ### Author Response · Authors · 2024-11-22
> > **References**
> >
> > - [1] [Nicoli, Kim A., et al. "Asymptotically unbiased estimation of physical observables with neural samplers." Physical Review E 101.2 (2020): 023304.](https://journals.aps.org/pre/abstract/10.1103/PhysRevE.101.023304)

---

> ### Comment · Reviewer_pv8Q · 2024-11-25
> **response to the rebuttal**
>
> I appreciate the authors' detailed response, which clarified some aspects of the submission. I have a few follow-up questions:
>
> - The authors mentioned a novel "physics-informed training procedure" in their response. Could the authors further clarify what "physics-informed" entails? Does it refer to incorporating physical constraints, governing equations, or other principles? I could not find related details in the manuscript or the rebuttal.
>
> - Could the authors explain how a trained $2 \times 2$ model is used to initialize a $4 \times 4$ model?
>
> - I have not yet seen the updated revision of the manuscript. I will reconsider my score once the updated version is available.

---

> > ### Author Response · Authors · 2024-11-28
> > **Response to follow up question of reviewer pv8Q**
> >
> > > I appreciate the authors' detailed response, which clarified some aspects of the submission. I have a few follow-up questions:
> >
> > We appreciate the reviewer’s engagement in the discussion.  Below, we answer the follow-up questions.
> >
> > > The authors mentioned a novel "physics-informed training procedure" in their response. Could the authors further clarify what "physics-informed" entails? Does it refer to incorporating physical constraints, governing equations, or other principles? I could not find related details in the manuscript or the rebuttal.
> >
> > By “physic-informed training procedure”, we mean that our “warm start with model transfer” strategy relies on scale invariance at criticality (SIC), which emerges within the theoretical framework of the Renormalization Group (RG) in physics. Therefore, the warm start procedure is, in this sense, “physics-informed”.  We added Fig. 2 in Section 3.3 of the revision (now uploaded!), where for each sequential training level, we show which RiGCS model (with its parameters)  is trained on which target Boltzmann distribution. Furthermore, the figure clarifies which parameters are used for initialization (warm-start) of the parameters in the following level.   The red arrows indicate the warm start initializations and the parameters connected by the arrows must be similar for our sequential training to be effective.  This is where the physical knowledge that (approximate) SIC emerges plays a central role.  Namely, the parameters connected by the red arrows are similar to each other under SIC.  In the revision, this explanation is further elaborated in the third paragraph of Appendix E.1.
> >
> > > Could the authors explain how a trained  model is used to initialize a  model?
> >
> > Fig.2 in the revision explicitly shows which parameters are initialized to which parameters, trained in the previous steps.
> >
> > > I have not yet seen the updated revision of the manuscript. I will reconsider my score once the updated version is available.
> >
> > The revision has now been uploaded.  Further questions/suggestions would be very welcome.

---

> > > ### Comment · Reviewer_pv8Q · 2024-11-29
> > > **response to the rebuttal**
> > >
> > > I appreciate the detailed response. My concerns are addressed, and I have adjusted my score accordingly.

---

> > > > ### Author Response · Authors · 2024-11-29
> > > >
> > > > Thank you very much for your review and discussion, which improved the readability of the paper significantly!

---

### Official Review · Reviewer_1WHN · 2024-10-27

**Soundness:** 2
**Presentation:** 3
**Contribution:** 3
**Rating:** 6
**Confidence:** 3

**Summary:**

In this paper, the authors present RiGCS, a method to sample from Boltzmann distributions in systems with critical phenomena such as phase transitions. This is challenging due to the problem of scale invariance at criticality, in which the correlation length between sites on a lattice diverges. This leads to a high rejection rate in standard Markov Chain Monte Carlo (MCMC) algorithms which mainly focus on local correlations to propose new configurations. The authors also state that this poses a challenge for generative models due to the need for large receptive fields. The proposed approach breaks down the system lattice into a hierarchy of resolutions, essentially factoring the probability distribution at the finest level into a product of distributions each conditioned on a coarser level. Each distribution is parameterized by autoregressive neural networks (PixelCNN) which are learned sequentially, and whose parameters are used to warm-start the next level in the hierarchy. The authors demonstrate that their method is able to generate samples which estimate free energies more accurately than the baselines, as well as have a higher effective sample size due to lower correlation between samples.

**Strengths:**

The paper tackles an important and challenging problem in physics (critical phenomena/phase transitions), and one that has received comparitively less attention from the AI for Science community, which is largely still focused on equilibrium phenomena. Thus, I think the topic/paper would be of interest to many in the community. Additionally, the proposed approach (hierarchical sampling at multiple resolutions) is interesting and solidly grounded in well-known techniques from scientific computing such as multigrid algorithms.

**Weaknesses:**

My primary concerns center around the "receptive field" argument, and the quantity and quality of the experimental results. I will elaborate further.

Receptive field:

There is a large and growing body of literature on long-context models in the fields of language and computer vision [1, 2, 3, 4, 5]. The goal of these works is to provide models with more global reasoning capabilities over, e.g. extremely long textual or video contexts, and/or to reason over extremely high-resolution images, while remaining memory and compute-efficient. The central problem tackled by this paper - diverging correlation lengths in critical systems require large receptive fields - is related to this line of work. The proposed solution - a hierarchical decomposition of the lattice - is also related to the scale hierarchies proposed in these works.

My first criticism is that the authors do not mention these works or position their contribution in the backdrop of this existing work. Can the authors comment on how their problem setting is substantively different from what is tackled in these existing works, and/or why the approaches from this line of work cannot be applied "zero-shot" to this application domain? My next and primary criticism is that given the current state-of-the-art in long context generative modeling, the proposed method is behind the curve, and the size of systems considered in this work (at most $128^2$) is not a sufficiently challenging setting to warrant the proposed approach. First of all, the authors use convolutional neural networks, which have the inherent limitation of a local receptive field. Can the authors justify this choice? Transformers have a global receptive field at the expense of an increased computational/memory footprint, but for 128x128 systems this shouldn't pose an issue. As far as I could tell, there was no mention of the Transformer architecture in the paper. Even if you consider much larger lattices (~$100^4$) used in e.g. lattice gauge theories and quantum chromodynamics, this is still comparable to the number of pixels in, e.g. satellite imagery data, for which solutions to reason over long context have been proposed [5]. I understand that there is not a one-to-one mapping between modeling critical phenomena and long context image/video models, but I think this disparity is telling and worth pointing out.

The choice to use an autoregressive model (PixelCNN) for sampling individual states also seems dubious to me. There is nothing inherently causal about the lattice sites, and so it seems it would make more sense to employ a non-autoregressive generative paradigm, such as diffusion. By parameterizing a diffusion model with e.g. a Transformer with bidirectional/unmasked attention, a global receptive field is naturally obtained at every stage of the generation process. I understand the binary nature of spin systems poses a slight hurdle, but one that is seemingly surmountable given the growing body of work on discrete diffusion [6] (not to mention more naive solutions like quantizing continuous predictions, clipping, binning, etc.). Plus, the authors mention that their approach is valid for continuous systems, so this seems like a natural choice (non autoregressive approaches are the method of choice for continuous data modalities).

Quantity/quality of experimental results:

Given my previous comments, I feel that the amount/nature of the experimental validation of the proposed method is too narrow. The evaluation is restricted to 2D Ising systems with size at most $128^2$. Can the authors include results on larger systems where, e.g. using a global receptive field is actually out of the question due to the memory constraints?

Additionally, the improvements in free energy estimation and EMDM shown in Figure 2 and 3 appear to be very marginal. The cluster method baseline significantly outperforms RiGCS on effective sample size - the authors claim that it is not applicable to continuous settings, but they do not demonstrate results in these settings either.

In summary, in order to reconsider my score, I would like to see considerable evidence that the proposed approach is superior to the many existing ways to achieve global receptive fields in generative models, including empirically on larger and continuous systems.

[1] Child, Rewon, et al. "Generating long sequences with sparse transformers." arXiv preprint arXiv:1904.10509 (2019).

[2] Liu, Ze, et al. "Swin transformer: Hierarchical vision transformer using shifted windows." Proceedings of the IEEE/CVF international conference on computer vision. 2021.

[3] Dao, Tri, et al. "Flashattention: Fast and memory-efficient exact attention with io-awareness." Advances in Neural Information Processing Systems 35 (2022): 16344-16359.

[4] Liu, Hao, Matei Zaharia, and Pieter Abbeel. "Ring attention with blockwise transformers for near-infinite context." arXiv preprint arXiv:2310.01889 (2023).

[5] Gupta, Ritwik, et al. "xT: Nested Tokenization for Larger Context in Large Images." arXiv preprint arXiv:2403.01915 (2024).

[6] Austin, Jacob, et al. "Structured denoising diffusion models in discrete state-spaces." Advances in Neural Information Processing Systems 34 (2021): 17981-17993.

**Questions:**

1. Presenting the proposed method as an alternative to MCMC sampling feels confusing to me. Doesn't the proposed method generate i.i.d samples from the Boltzmann distribution? As opposed to MCMC approaches which produce correlated samples via a Markov Chain? Unless I am misunderstanding something, I think it would help to clarify this point.

2. How is the training data obtained for optimizing the KL divergence loss? Is it from long MCMC simulations? I think this is a critical point that was not focused upon enough in the paper, unless I missed something.

---

> ### Author Response · Authors · 2024-11-22
> **Reply to Reviewer 1WHN 1/2**
>
> We thank the referee for their comments, which informed us that we did not explicitly outline the crucial requirements for generative models in the context of our target problem—Monte Carlo simulations for estimating physical observables.  First of all, we have an important correction to the experimental results in Fig.2 (left).  Please see our general reply and also note that we have not uploaded a revision yet, but we plan on doing it before the discussion phase ends.
>
> > My first criticism is that the authors do not mention these works or position their contribution in the backdrop of this existing work. Can the authors comment on how their problem setting is substantively different from what is tackled in these existing works, and/or why the approaches from this line of work cannot be applied "zero-shot" to this application domain?
>
>  As the referee suspected, there is a fundamental difference in the problem setting from the typical applications in computer vision and natural language processing, which prevents us from using the state-of-the-art architectures (e.g., transformers, GANs, diffusion models) for general machine learning applications. The reasons are twofold.
>
> - 1. **The exact sampling probability needs to be efficiently accessible**: When estimating physical observables from Monte Carlo (MC) simulations, one estimates both the mean value and the uncertainty, which is crucial to provide confidence intervals, e.g.,  for validating theory (e.g., the standard model) with experiments (e.g, high energy physics experiments in a collider).  The confidence interval is estimated by making the estimators unbiased, and evaluating their variances (the variance is easily evaluated, while the bias is not unless the true value is known).  Unlike MCMC methods, which asymptotically provide samples *exactly* from the target distribution, generative neural samplers (GNSs) can **not** directly sample from the exact target distribution, which makes the MC estimator of GNSs  biased.   As mentioned in the first paragraph of Section 2.2, one can make the estimator unbiased by re-weighting (Neural Importance sampling or NeuralMCMC [1]) **if and only if** the sampling probability is **accessible**.  For this reason, modern research in the field focuses on using generative models with accessible sampling probabilities, such as autoregressive models, normalizing flows, and variants thereof, which are standard choices for sampling discrete and continuous variables.   Diffusion models are actually potential candidates because the stochastic differential equation (SDE) for the reverse diffusion has an equivalent ordinary differential equation (ODE) (i.e., probability flow, continuous flow, or neural ODE).  However, the SDE is solved approximately, and an efficient method to compute the exact sampling probability is yet to be established, to the best of our knowledge.
>
> - 2. **Training needs to be done without a large number of samples from the target distribution**: Another crucial difference from the more typical ML problems is that we cannot assume the existence of the training data drawn from the target distribution.  If a large amount of data from the target distribution, on which the state-of-the-art architectures like transformers with large receptive fields can be trained,  are available, one can simply compute the MC estimator for the observables without training a generative model.  Although one could assume some situations where sufficiently large training data for a few physical systems, e.g., Ising models for a few coupling parameter values, are available, and a conditional generative model is trained only on those data to generalize it for many other parameter settings, this is not a common scenario and indeed is not our setting. We refer to a recent review paper, and references therein, where the problem of using generative models for Lattice QCD is tackled [1].
>
> Because of these two requirements, state-of-the-art ML models like transformers with large receptive fields cannot be used for our target application. Following previous state-of-the-art generative neural samplers for the Ising model, VAN [2] and HAN [3], we adopted the PixelCNN architecture as the basic component of our generative model.  Consequently, we need to limit the receptive field size of the PixelCNN for tractability.  As shown in Fig. 2 (left), the plain VAN with receptive fields containing all lattice sites can be trained only up to systems of size $32 \times 32$ (training time explodes to several weeks/months for larger systems).
>
> We will make these requirements explicit in the revision.

---

> > ### Author Response · Authors · 2024-11-22
> > **Reply to Reviewer 1WHN 2/2**
> >
> > > Given my previous comments, I feel that the amount/nature of the experimental validation of the proposed method is too narrow. The evaluation is restricted to 2D Ising systems with size at most $128^2$. Can the authors include results on larger systems where, e.g. using a global receptive field is actually out of the question due to the memory constraints?
> >
> > Because of the two requirements above, the range of suitable architectures is limited. Sampling lattices of size $128^2$ is a very hard problem for the generative samples fulfilling the requirements above.  We note that in [3], the previous state-of-the-art generative neural sampler failed to obtain good results for $128\times 128$ lattices, thus making RiGCS the new state-of-the-art in the field.
> >
> > > Additionally, the improvements in free energy estimation and EMDM shown in Figure 2 and 3 appear to be very marginal. The cluster method baseline significantly outperforms RiGCS on effective sample size - the authors claim that it is not applicable to continuous settings, but they do not demonstrate results in these settings either.
> >
> > Cluster methods and specialized MCMC methods (e.g., HMC), in general, are known to be strong baselines for most physical systems, and no existing works on generative models for simulating such systems have reported improved performance in estimating *non-thermodynamic* observables.  A few works [3,5,6] reported better performance in estimating thermodynamic observables, *similar to what we did* for the free energy in Fig. 2 left in the original submission (**note** that we will correct this result in the revision as mentioned in the general reply).  The domain of generative neural samplers for physical systems (sometimes also known as Boltzmann generators)  is an emerging field where the current state-of-the-art has not yet achieved the same level of performance offered by standard MCMC methods for large systems. Researchers continue to advance the state-of-the-art in “generative” neural samplers by introducing new ideas, often inspired by physical priors, like RiGCS. Such innovations are essential for developing viable alternatives to traditional MCMC methods.
> >
> > > In summary, in order to reconsider my score, I would like to see considerable evidence that the proposed approach is superior to the many existing ways to achieve global receptive fields in generative models, including empirically on larger and continuous systems.
> >
> > Within the possible generative neural samplers (GNSs) that satisfy the two fundamental requirements explained above, our method is arguably state-of-the-art, as shown in Figs. 2 and 3.
> >
> > > Q1: Presenting the proposed method as an alternative to MCMC sampling feels confusing to me. Doesn't the proposed method generate i.i.d samples from the Boltzmann distribution? As opposed to MCMC approaches which produce correlated samples via a Markov Chain? Unless I am misunderstanding something, I think it would help to clarify this point.
> >
> > As the referee mentioned, MCMC only generates correlated samples. This introduces two shortcomings.  First, correlated samples are harmful because they reduce the effective sample size. Second, MCMC does not give access to the sampling probability.  Generative models have the potential to offer a better alternative for MCMC samplers because they address both shortcomings. First, they can generate independent and identically distributed (iid) samples, and second, they give access to a tractable density, which is necessary for directly estimating some important physical observables, e.g., the free energy [5].
> >
> > > Q2: How is the training data obtained for optimizing the KL divergence loss? Is it from long MCMC simulations? I think this is a critical point that was not focused upon enough in the paper, unless I missed something.
> >
> > We train RiGCS by minimizing the “reverse” KL divergence, Eq. (11), which does not require samples from the target distribution.  We made this point clearer in the revised manuscript.

---

> > > ### Author Response · Authors · 2024-11-22
> > > **References**
> > >
> > > - [1] [Cranmer, Kyle, et al. "Advances in machine-learning-based sampling motivated by lattice quantum chromodynamics." Nature Reviews Physics 5.9 (2023): 526-535.](https://www.nature.com/articles/s42254-023-00616-w)
> > > - [2][Wu, Dian, Lei Wang, and Pan Zhang. "Solving statistical mechanics using variational autoregressive networks." Physical review letters 122.8 (2019): 080602.](https://journals.aps.org/prl/abstract/10.1103/PhysRevLett.122.080602)
> > > - [3] [Białas, Piotr, Piotr Korcyl, and Tomasz Stebel. "Hierarchical autoregressive neural networks for statistical systems." Computer Physics Communications 281 (2022): 108502.](https://www.sciencedirect.com/science/article/pii/S0010465522002211)
> > > - [4] [Richter, Lorenz, and Julius Berner. "Improved sampling via learned diffusions." ICLR (2024).](https://openreview.net/forum?id=h4pNROsO06)
> > > - [5] [Nicoli, Kim A., et al. "Estimation of thermodynamic observables in lattice field theories with deep generative models." Physical review letters 126.3 (2021): 032001.](https://journals.aps.org/prl/abstract/10.1103/PhysRevLett.126.032001)
> > > - [6] [Caselle, Michele, et al. "Stochastic normalizing flows as non-equilibrium transformations." Journal of High Energy Physics 2022.7 (2022): 1-31.](https://link.springer.com/article/10.1007/JHEP07(2022)015)

---

> > ### Comment · Reviewer_1WHN · 2024-11-22
> > **Response**
> >
> > Thanks to the authors for their response and clarification on requiring exact sampling probabilities. I'm still not entirely convinced by this argument, particularly point 1. A distinction should be drawn between **modeling frameworks** and **architectures**. Diffusion models, GANs, autoregressive models, and normalizing flows are modeling frameworks, while CNNs, transformers, etc. are architectures which parameterize the modeling framework. The accessibility of the exact sampling probability is a function of the modeling framework, while the receptive field is a function of the architecture. Therefore, it should be possible replace a PixelCNN with a transformer, holding the modeling framework constant, and obtain a global receptive field while still computing exact sampling probabilities, right? If so, then the argument for not using existing receptive field expansion techniques boils down to point 2, which, if I interpreted it correctly, is saying that transformers are too data-hungry to be used in limited data settings such as this one. I'm not satisfied with the authors simply asserting this; I would like to see empirical evidence that such global receptive methods do not learn well with the quantity of data available.
> >
> > Please let me know if I am still misunderstanding something. I will wait to update my score until I see the updated paper.

---

> ### Author Response · Authors · 2024-11-28
> **Response to follow up question of reviewer 1WHN**
>
> We have now uploaded a revision.  We appreciate the reviewer’s follow-up questions, giving us a further chance to convince them.  We agree that the model and the architecture can be distinguished, and we understand the reviewer’s request to evaluate the VAN model with a transformer architecture.
>
> We explored three possibilities to conduct the requested experiment, which could be finished by the deadline.
> - 1) Looking for existing methods with their code available and testing one of them.
> - 2) Implement a VAN with a transformer by ourselves.
> - 3) Test our RiGCS with a transformer as the basic building block.
>
> For Option 1, we found Transformer Neural Autoregressive Flows [1], which we could try to train with self-generated samples (via the reverse KL divergence) or with limited numbers of training data drawn by MCMC methods from the target Boltzmann distribution (via the forward KL divergence or the maximum likelihood training).  However, the paper has not been published in conference nor journal (although presented in a workshop), and therefore they have not pushed their code into their GitHub repository.  We further explored the literature, but so far we could not find a suitable existing method.  We are happy to try this option by the (extended!) deadline if the reviewer would suggest an existing method with the code publicly available, such that we can easily test its performance.
>
> For Option 2, we considered if there would be an easy way to implement VAN with a transformer, e.g., by replacing the PixelCNN with a transformer in the VAN implementation [2].  However, we found that it is not straightforward as we would need to carefully design a transformer-based layer that can deal with masked weights.  A priori, it is hard to determine whether it is more efficient to use the standard encoder-decoder transformer architecture or to use transformer layers with masked weights like in the MADE model. Implementing this approach would require substantial time and rigorous testing, making it unrealistic to complete within the limited timeframe of the rebuttal period.
>
> Option 3 is viable, as in the conditional models for RiGCS one masks the lattices rather than the weights, as usually done with standard autoregressive models like MADE or PixelCNN. We thus replaced CNNs with transformer layers and tested them by keeping the PixelCNN at the coarsest level $l=0$ (because it is the plain VAN).  The result for $16 \times 16$ lattice can be found at the [**ANONYMIZED LINK**](https://www.dropbox.com/scl/fi/mhbjin8biun8eucs2s1jh/TransformerVsCNNbinned16x16.pdf?rlkey=xgjkqx8kvnizh4z65nykrre4s&st=taia6cdu&dl=0), where RiGCS with Transformer (blue) and RiGCS with CNN (orange) are compared in terms of ESS during training.  Note that the plot also shows the pre-training phase, i.e., the first phase (1-1001 epochs) corresponds to the pretraining of unconditional VAN at $l=0$ (where RiGCS with transformer also uses CNN and thus the performance is comparable). The second (1002-11002 epochs) and the third (11003-21003 epochs) phases involve training of the conditional VANs at $l=1, 2$, and the last phase (21004- epochs) corresponds to the training of the full RiGCS after pre-training for $N=16$.  We clearly see that RiGCS with CNN outperforms RiGCS with Transformer.  Note that the ESS shown in the figure is for $2 \times 2$, $4 \times 4$, $8 \times 8$, and $16 \times 16$, respectively, in the first, second, third pretraining phases, and the last full training phase.
>
> We understand that Option 3 is not exactly what the reviewer requested.  However, we found it to be the only feasible implementation given the time-constraints of the rebuttal period.  More importantly, our result is consistent with the observed preference for CNNs over transformers in the field of generative neural samplers (GNSs).  For example, in  [3], where GNSs are used to simulate lattice QCD, the authors concluded in the last paragraph of Section V-A that *“[...] While the attention mechanism is at the heart of some of the most powerful artificial intelligence applications to date, we find that in our experiments, convolution architectures generally train faster and result in higher performances. [...]”*.  Considering that negative results are less likely to be published, the current preference for CNNs over transformers in lattice sampling is also evident from the fact that public implementations of GNSs using transformers are rare (as seen in our challenges with Option 1), despite that  CNN-based normalizing flows and ARNNs are widely available.
>
> While we may not fully address the reviewer’s request, we kindly ask for consideration of the practical constraints (Options 1 and 2 are challenging within a short timeframe) and the prevailing consensus in the GNS field that CNNs are currently the more practical and effective choice of architecture.

---

> > ### Author Response · Authors · 2024-11-28
> > **Refs**
> >
> > - [1] [Massimiliano Patacchiola et al. "Transformer Neural Autoregressive Flows" arXiv preprint arXiv:2401.01855 (2024).](https://arxiv.org/abs/2401.01855)
> > - [2]  [Wu, Dian, Lei Wang, and Pan Zhang. "Solving statistical mechanics using variational autoregressive networks." Physical review letters 122.8 (2019): 080602.](https://journals.aps.org/prl/abstract/10.1103/PhysRevLett.122.080602)
> > - [3] [Abbott, Ryan et al. "Normalizing flows for lattice gauge theory in arbitrary space-time dimension." arXiv preprint arXiv:2305.02402 (2023).](https://arxiv.org/pdf/2305.02402)

---

> > > ### Comment · Reviewer_1WHN · 2024-12-01
> > > **Response**
> > >
> > > I appreciate the authors doing their best to address my feedback regarding the receptive field/transformer aspects of the work. Option 3, which the authors have implemented, is actually roughly what I had in mind when I requested the additional experiment. I also appreciate the pointers to literature discussing that CNNs seem to outperform transformers for this problem. Given these improvements, as well as those made in response to the other reviewers' feedback, I am willing to raise my score to a 6. I think this is a thoughtful paper which tackles an interesting problem, and would be of interest to the ICLR audience.
> > >
> > > I understand that the period for authors to modify the paper has ended, but if accepted, I would like to see a discussion of the transformer/receptive field experiments/tradeoffs in the final text. I think it is important to highlight the challenges associated with using a Transformer in this setting, and possible paths forward.

---

> > > > ### Author Response · Authors · 2024-12-01
> > > >
> > > > Thank you for your discussion and reconsideration of the score.  We will add the discussion on the transformer experiments in the final version.

---

### Official Review · Reviewer_cr5x · 2024-10-31

**Soundness:** 3
**Presentation:** 3
**Contribution:** 3
**Rating:** 6
**Confidence:** 4

**Summary:**

This paper deals with the problem of sampling systems at criticality, i.e. systems typically displaying some form of self-similarity w.r.t change of scale, and estimating their associated thermodynamical quantities like the free energy. This is a notoriously difficult problem of interest in physics, because it requires to deal with arbitrary long range correlations and for which standard MC samplers based on local moves undergo-critical slowing down, meaning that the auto-correlation time become arbitrarily large with system size. As explained in the introduction, there exists some specific algorithms in the physics literature, like cluster methods specifically designed for lattice models,
typically working for discrete or continuous variables (Ising, Potts, O(n) ...). In this work, the authors propose to leverage machine learning methods to generalized some existing multi-scale methods following closely the spatial RG procedures, which basically exploits statistical dependencies between coarse grained variables at different scales. These use rather drastic approximations on the conditional models between coarse grained variables at different scales, leading sometime to very low acceptance rate, and ML methods offer the possibility to used more complex conditional models between scales.

**Strengths:**

the paper is well written and rather clear with an impressive bibliography on the subject which is indeed quite vast and old. A synthetic and pedagogical section in the supplementary material which should be useful to non-physicists is devoted to the RG theory, and should make the whole paper readable for non-physicists. The proposed algorithm is quite generic and and is competitive with the Wolff algorithm on 2d lattices up to L=128, where it seems to actually yield better estimates of the free energy. and shows clear advantage w.r.t MLMC-HB (used in it orginal version maybe outdated, as it use the original  Kadanov block-spin  transformation, subsequent refined algorithms cited in the text are not considered for  baselines) and other baselines considered like HAN which uses a similar idea, in the form of an auto-regressive
NN based on a hierarchical representation of the lattice.

**Weaknesses:**

I found some aspects of the experimental not completely satisfactory. The claim that RiGCS performs better than Wolff is a strong claim that should be better sustained in my opinion. Authors concentrate in the experiments on the computation of the free energy, but characterization of physical properties involve many other possible indicators, which the  Wolff algorithm is known to directly compute very efficiently,
like the magnetization, susceptibility, Binder parameters which could be easily displayed without further work it seems ...
some exponents could also be extracted by finite size scaling and compared to exact ones. The combination of Wolff with AIS even though reasonable seems a bit like of an ad-hoc procedure to yield a baseline comparison for the computation of the free energy, I would expect that integrating the mean energy with respect to beta or simply using directly the empirical distribution of energy delivered by the Wolff algorithm would yield more robust estimation but I maybe wrong? The size itself (128x128) seems rather limited compared to what can commonly be achieved with the Wolff algorithm. It seems to be a limitation in term of computation time but this is not discussed. Actually the scaling of the the computational time  of RiGCS w.r.t to L is actually not given. Another possible weak  point is that the proposal of the paper is very similar in spirit with another recent one which also uses a top-down regressive model  in scales (also explicitly using  RG concepts), in combination with wavelets (Marchand et al. 2022). While this work is mentioned in the paper, it is not really discussed, no element of comparison or even qualitative elements describing advantages of  RiGCS over this one are being given.  Finally the experiments concentrate on 2D Ising paradigm, a larger spectrum of experiments, on models with continuous variables in particular, would be beneficial to convince the reader of the flexibility of the method.

Despite these reserves, I find the method quite appealing due to its simplicity and the performances which are displayed.

**Questions:**

If the author could comment on the points raised in the previous paragraph:
in particular
- How does the algorithm scale  system size (even empirically) ?
- Is there a key difference of RIGCS wrt   the model developed by Marchand et al. (Wavelet conditional renormalization group)?


Minor points:

Instead of what is written on p4, the Wolff algorithm seems also to work for continuous variables like O(n) variables as can be seen  for instance in Kent-Dobia, Sethna PRE (2018)

I had difficulties to understand the notations in the begining, concerning in particular the definition of the hierarchy of variables.  I am not sure how I should understand the last sentence of p4 for instance, whether (i) there are coarse grained variables which are defined hierarchically or whether (ii) the original variables are organized hierarchically depending on their position on the lattice as in the Schmidt paper (I believe it is (ii) after reading it)

bibliography could be ordered in alphabetic order (to help old school readers using printed version:) )

---

> ### Author Response · Authors · 2024-11-22
> **Reply to Reviewer cr5x 1/2**
>
> We thank the referee for raising important points and for their thoughtful questions.   We have an important correction to the experimental results in Fig.2 (left).  Please see our general reply and also note that we have not uploaded a revision yet, but we plan on doing it before the discussion phase ends.
> Below we reply to the reviewer’s enquiries.
>
> > Weakness: I found some aspects of the experimental not completely satisfactory. The claim that RiGCS performs better than Wolff is a strong claim that should be better sustained in my opinion. Authors concentrate in the experiments on the computation of the free energy, but characterization of physical properties involve many other possible indicators, which the Wolff algorithm is known to directly compute very efficiently, like the magnetization, susceptibility [...].
>
> In the original submission, we claimed that RiGCS outperforms the state-of-the-art generative model baselines (in the abstract), and that it outperforms the cluster method *in free energy estimation (Line 402)*. Note that this claim is not as strong as claiming that RiGCS performs *better than the Wolff algorithm in general*. However, we need to modify our claims because of the error reported in the general reply.  In the revision, we do not claim that RiGCS outperforms the cluster method in *any* of our experiments.  As the referee mentioned, the cluster method is a very strong sampler for the Ising model, and no generative neural sampler has been able to outperform it.  In the revised manuscript, we will make it clearer that our RiGCS outperforms **only** the state-of-the-art **generative** samplers.
>
>  > The combination of Wolff with AIS even though reasonable seems a bit like of an ad-hoc procedure to yield a baseline comparison for the computation of the free energy, I would expect that integrating the mean energy with respect to beta or simply using directly the empirical distribution of energy delivered by the Wolff algorithm would yield more robust estimation but I maybe wrong?
>
> We chose Cluster-AIS as the state-of-the-art method for free energy computation with MCMC samplers.  Recent works [1-3] showed that AIS combined with Cluster/MCMC significantly outperforms the integral method, where fine discretization of the temperature parameter is required to ensure overlapping distribution supports between the subsequent steps. We do not know how to compute the free energy directly from the “empirical distribution of energy” using the Wolff algorithm.  We would appreciate it if the referee could provide more explanations or references about it.
>
> > The size itself (128x128) seems rather limited compared to what can commonly be achieved with the Wolff algorithm. It seems to be a limitation in term of computation time but this is not discussed. Actually the scaling of the the computational time of RiGCS w.r.t to L is actually not given.
> > Question: How does the algorithm scale system size (even empirically) ?
>
> We agree that $L=128$ is limited compared to the capability of the Wolff algorithm.  In the revision, we made explicit that we claim state-of-the-art performance only with respect to the generative neural sampler (GNS) baselines, i.e., VAN and HAN.  In the context of GNSs, sizes of $128\times128$  already represent a large and challenging problem in comparison to other recent works [4-6].
>
> Leveraging physical information, i.e., scale invariance at criticality (SIC), we designed RiGCS with multilevel conditional samplers, where the conditional samplers are convolutions with a fixed filter size (receptive field) for all levels.  This allows linear model complexity (the number of network parameters) with respect to the number of levels, e.g., complexity scales linearly with the lattice sites $(V = L^2)$.  We will add this argument and empirical computation time, which verifies the linear scaling, in the revised manuscript.

---

> > ### Author Response · Authors · 2024-11-22
> > **Reply to Reviewer cr5x 2/2**
> >
> > > Another possible weak point is that the proposal of the paper is very similar in spirit with another recent one which also uses a top-down regressive model in scales (also explicitly using RG concepts), in combination with wavelets (Marchand et al. 2022). While this work is mentioned in the paper, it is not really discussed, no element of comparison or even qualitative elements describing advantages of RiGCS over this one are being given.
> > > Question: Is there a key difference of RIGCS wrt the model developed by Marchand et al. (Wavelet conditional renormalization group)?
> >
> > We did not discuss [7] in the main text because the problem setting is completely different.  The task tackled in [7] is to learn the energy (Hamiltonian) from a large number of observed samples. In contrast, our task is to train a generative sampler that approximates the target Boltzmann distribution specified by a given energy function without **any** training data from the target distribution.  In the revised manuscript, we will describe our problem setting more clearly. We will also discuss the differences between RiGCS and [7] in the extended related work in Appendix A, where we introduced weakly related works, e.g., with a greater focus on applications of the renormalization group to machine learning.
> >
> > > Finally the experiments concentrate on 2D Ising paradigm, a larger spectrum of experiments, on models with continuous variables in particular, would be beneficial to convince the reader of the flexibility of the method.
> >
> > We are implementing new theories to benchmark our method.  We plan to update the revision with new results before the discussion phase ends (see also general rebuttal).
> >
> > > Minor points
> > > Instead of what is written on p4, the Wolff algorithm seems also to work for continuous variables like O(n) variables as can be seen for instance in Kent-Dobia, Sethna PRE (2018)
> >
> > We thank the referee for bringing the reference Kent-Dobia, Sethna PRE (2018) to our attention. We revised our claim on page 4 taking this reference into account.
> >
> > > I had difficulties to understand the notations in the beginning, concerning in particular the definition of the hierarchy of variables. I am not sure how I should understand the last sentence of p4 for instance, whether (i) there are coarse grained variables which are defined hierarchically or whether (ii) the original variables are organized hierarchically depending on their position on the lattice as in the Schmidt paper (I believe it is (ii) after reading it)
> >
> > The intuition (ii) of the reviewer is correct.  We added a note in the revised manuscript to make sure that readers do not misunderstand.
> >
> > > bibliography could be ordered in alphabetic order (to help old school readers using printed version:) )
> >
> > The bibliography is ordered in the alphabetic order **by the last name of the first author**.
> >
> > We hope our reply addressed all the concerns and questions raised by the referee. Should there be any additional inquiries, we would be pleased to continue the discussion.

---

> > > ### Author Response · Authors · 2024-11-22
> > > **References**
> > >
> > > - [1] [Caselle, Michele, et al. "Jarzynski’s theorem for lattice gauge theory." Physical Review D 94.3 (2016): 034503.](https://journals.aps.org/prd/abstract/10.1103/PhysRevD.94.034503)
> > > - [2] [Caselle, Michele, Alessandro Nada, and Marco Panero. "QCD thermodynamics from lattice calculations with nonequilibrium methods: The SU (3) equation of state." Physical Review D 98.5 (2018): 054513.](https://journals.aps.org/prd/abstract/10.1103/PhysRevD.98.054513)
> > > - [3] [Bulgarelli, Andrea, and Marco Panero. "Entanglement entropy from non-equilibrium Monte Carlo simulations." Journal of High Energy Physics 2023.6 (2023): 1-33.](https://link.springer.com/article/10.1007/jhep06(2023)030)
> > > - [4] [Li, Shuo-Hui, and Lei Wang. "Neural network renormalization group." Physical review letters 121.26 (2018): 260601.](https://link.aps.org/doi/10.1103/PhysRevLett.121.260601)
> > > - [5][Wu, Dian, Lei Wang, and Pan Zhang. "Solving statistical mechanics using variational autoregressive networks." Physical review letters 122.8 (2019): 080602.](https://journals.aps.org/prl/abstract/10.1103/PhysRevLett.122.080602)
> > > - [6] [Białas, Piotr, Piotr Korcyl, and Tomasz Stebel. "Hierarchical autoregressive neural networks for statistical systems." Computer Physics Communications 281 (2022): 108502.](https://www.sciencedirect.com/science/article/pii/S0010465522002211)
> > > - [7] [Marchand, Tanguy, et al. "Multiscale data-driven energy estimation and generation." Physical Review X 13.4 (2023): 041038.](https://journals.aps.org/prx/abstract/10.1103/PhysRevX.13.041038)

---

> > > > ### Comment · Reviewer_cr5x · 2024-11-25
> > > >
> > > > I thanks the authors for answering my points of concern  especially the one regarding the situation w.r.t. Wolff algorithm. After weighting the pros and cons arguments I think this work is worth publishing at ICLR so I raise my score.

---

> > > > > ### Author Response · Authors · 2024-11-28
> > > > > **Revision upload**
> > > > >
> > > > > We thank the reviewer for the active communication and reconsideration of the score.  We have now uploaded a revision, where most of the reviewer’s comments have been incorporated (except for some experimental parts).  Further suggestions to improve the paper would be very much appreciated.

---

### Official Review · Reviewer_DmX4 · 2024-11-04

**Soundness:** 4
**Presentation:** 3
**Contribution:** 4
**Rating:** 8
**Confidence:** 2

**Summary:**

The paper presents a novel approach, the Renormalization-informed Generative Critical Sampler (RiGCS), designed for efficient sampling in near-critical regimes, leveraging the concept of scale invariance at criticality (SIC). By integrating MultiLevel Monte Carlo (MLMC) with Heat Bath (HB) algorithms, the authors aim to improve sampling efficiency while addressing challenges associated with critical slowing down in Markov chain Monte Carlo (MCMC) methods.

**Strengths:**

The proposed method utilizes SIC effectively, enabling ancestral sampling from low-resolution to high-resolution lattice configurations with site-wise-independent conditional HB sampling. This innovation enhances both scalability and efficiency, making it a significant contribution to the field of generative models.

**Weaknesses:**

(1) It appears that the training of the algorithm at each resolution follows a simultaneous sampling and training paradigm. However, this is not clearly articulated in the manuscript. I recommend providing complete pseudocode for the algorithm, which would greatly aid readers in understanding the implementation and workflow.

(2) The loss function presented in equation (11) is known to have a mode-seeking tendency when training generative models. This could lead to the neglect of certain peaks in multimodal distributions. It would be beneficial for the authors to address whether RiGCS is susceptible to similar issues and, if so, how they might mitigate them.

(3) The paper does not mention an important class of methods for the Ising model that rely on relaxation to continuous variable representations. Approaches such as the Neural Network Renormalization Group by Li and Wang use this relaxation technique to enable generative sampling in a continuous variable space. Including a discussion of these relaxation-based methods would provide a more comprehensive comparison and allow readers to better appreciate where RiGCS stands in relation to existing methods.

**Questions:**

(1) Provide a detailed algorithm or flowchart that outlines the key steps of RiGCS, including how training proceeds at each resolution level. This would help clarify the simultaneous sampling and training processes.

(2) Address whether RiGCS is susceptible to the mode-seeking, if so, how they might mitigate them.

(3) Compare the proposed approach to methods like the Neural Network Renormalization Group, highlighting potential advantages or trade-offs of RiGCS in relation to these continuous relaxation based techniques.

---

> ### Author Response · Authors · 2024-11-22
> **Reply to Reviewer DmX4**
>
> We thank the reviewer for their encouraging comments and their thoughtful questions.  We have an important correction to the experimental results in Fig.2 (left),  please see our general reply.  Please also note that we have not uploaded a revision yet, but we plan on doing it before the discussion phase ends.
>
> First, we thank the reviewer for the positive assessment of our work.
> > Strengths:
> > The proposed method utilizes SIC effectively, enabling ancestral sampling from low-resolution to high-resolution lattice configurations with site-wise-independent conditional HB sampling. This innovation enhances both scalability and efficiency, making it a significant contribution to the field of generative models.
>
> Below, we reply to the reviewer’s inquiries.
>
> > Weaknesses (1): It appears that the training of the algorithm at each resolution follows a simultaneous sampling and training paradigm. However, this is not clearly articulated in the manuscript. I recommend providing complete pseudocode for the algorithm, which would greatly aid readers in understanding the implementation and workflow.
>
> > Question (1): Provide a detailed algorithm or flowchart that outlines the key steps of RiGCS, including how training proceeds at each resolution level. This would help clarify the simultaneous sampling and training processes.
>
> In the revised manuscript, which we will upload by the end of the discussion phase, we will include a figure explaining the training workflow in the main text, as well as a pseudocode in the appendix F.6.
>
> > Weaknesses (2): The loss function presented in equation (11) is known to have a mode-seeking tendency when training generative models. This could lead to the neglect of certain peaks in multimodal distributions. It would be beneficial for the authors to address whether RiGCS is susceptible to similar issues and, if so, how they might mitigate them.
> > Question (2) Address whether RiGCS is susceptible to the mode-seeking, if so, how they might mitigate them.
>
> We agree that the mode-seeking tendency is a concern.  In the manuscript we evaluated the effective mode-dropping measure (EDMD), a metric with which the bias caused by mode-dropping can be bounded [1].  We observe in Fig. 3 left that RiGCS is hardly prone to mode-dropping, unlike  the baseline GNSs (VAN, HAN). This result is consistent with the free energy estimation (Fig.2 left).  When RiGCS would suffer from the mode-dropping, standard remedies from the literature, such as annealing or forward-KL training [1], should be applied to mitigate these effects.
>
> > Weaknesses (3): The paper does not mention an important class of methods for the Ising model that rely on relaxation to continuous variable representations. Approaches such as the Neural Network Renormalization Group by Li and Wang use this relaxation technique to enable generative sampling in a continuous variable space. Including a discussion of these relaxation-based methods would provide a more comprehensive comparison and allow readers to better appreciate where RiGCS stands in relation to existing
> methods.
>
> > Question (3): Compare the proposed approach to methods like the Neural Network Renormalization Group, highlighting potential advantages or trade-offs of RiGCS in relation to these continuous relaxation based techniques.
>
> NeuralRG [2] aims to learn the renormalization operations with hierarchical bijective mapping, where the depth of the bijective layers depends on the level of the latent variables in the hierarchy.  Although this approach reduces the degrees of freedom of the model (i.e., the number of the network parameters) for efficiency, the reduction is not as drastic as for our RiGCS, because long-range correlations are still needed to be captured by the high-resolution layers.  In contrast, our RiGCS significantly limits the receptive field size to be *constant* for all layers, based on the function-forms of the renormalized Hamiltonians, leading to significant reduction of the model complexity.  As a quick evaluation, we tested the code for Neural RG [2] on a local GPU (using `device=mps` on M1 MacBookPro with 32G RAM).  We observed that training Neural RG takes 1h 30min for L=16, which is much slower than training RiGCS (which takes less than 15 min), supposedly due to its higher model complexity. We include NeuralRG [2]  and discuss the comparison with our method in the related work section of the revision.
>
>
> ## References
> - [1] [Nicoli, Kim A., et al. "Detecting and mitigating mode-collapse for flow-based sampling of lattice field theories." Physical Review D 108.11 (2023): 114501.](https://journals.aps.org/prd/abstract/10.1103/PhysRevD.108.114501)
> - [2] [Li, Shuo-Hui, and Lei Wang. "Neural network renormalization group." Physical review letters 121.26 (2018): 260601.](https://link.aps.org/doi/10.1103/PhysRevLett.121.260601)

---

> > ### Comment · Reviewer_DmX4 · 2024-11-24
> >
> > Thank authors for their detailed response. After careful consideration, I have decided to maintain my original score.

---

> > > ### Author Response · Authors · 2024-11-28
> > > **Revision upload**
> > >
> > > We thank the reviewer for the active communication.  We have now uploaded a revision, where most of the reviewer’s comments have been incorporated (except for some experimental parts).  Further suggestions to improve the paper would be very much appreciated.

---

### Author Response · Authors · 2024-11-22
**General Reply**

We thank the referees for insightful comments including a shared positive consensus regarding the key idea behind RiGCS as well as constructive criticisms. In this general reply, we first report on a correction of our experimental result, and then address the overarching concerns from multiple referees.  Note that we have not uploaded a revision yet as we are still working on incorporating all the suggestions. We will upload it before the discussion phase ends.

- **Error in Fig.2 left (free energy estimation result)**: First of all, we apologize that our experimental results in the original submission contained wrong information, specifically in Fig. 2 (left) where the free energy estimation was evaluated.  Originally, we observed that for L=64 and 128, cluster-AIS was biased, and we claimed that our RiGCS outperformed the cluster-AIS in this particular evaluation.  However, after careful investigations, we identified numerical problems both in the analytic solution computation (ground-truth) and the cluster-AIS implementation.  We replaced both values with the correct ones in the revised manuscript. In the new plot, both cluster-AIS and RiGCS give unbiased estimates, and cluster-AIS gives smaller variance.  We will adjust our claims accordingly, as explained below.

- **Comparison with the cluster MCMC methods**: In the field of sampling physical systems with generative neural samplers (GNSs), there is a general consensus that, for each physical system, there is a powerful taylored MCMC method, e.g., cluster methods for discrete variables including Ising models, and Hamiltonian MC (HMC) for continuous fields including lattice field theories. Those methods have hardly been outperformed by the current state-of-the-art GNSs. We refer to a recent review, and references therein, which discusses this problem in the context of Lattice QCD [1] as well as to refs in the original manuscript between lines 111-130.  A few exceptions, such as [2], reported better performance than HMC (using the integration method) in free energy estimation and [3] in mitigating topological freezing in Lattice QCD. In our original submission, we observed similar results to [2] and consequently claimed that our method outperformed cluster-AIS in free energy estimation; however, this claim was incorrect, for the reasons given in the point above.
 Therefore, we revised our claim to make it clear that our method **does not outperform** the cluster-AIS in *any* experiment.  Nevertheless, we argue that our work is a significant contribution to the field of GNSs, where new generative models with better scalability are extensively being developed with a hope that the advantages of GNS, i.e., independent sampling and direct partition function estimation, will be practically useful in the near future.  Our RiGCS is an attempt to improve the scalability by leveraging physical knowledge, achieving a linear model complexity with respect to the number of lattice sites with minimum approximation error.  We empirically demonstrated the improvement by showing its superior performance compared to VAN and HAN, its predecessor and the previous state-of-the-art GNS for the Ising model. We made these points clear in the revision.

- **Additional Experiments**: A significant concern raised by the referees pertains to the limited scope of the experiments, which focus exclusively on the two-dimensional Ising model. We chose this commonly used benchmark [4,5] not only because of its simplicity but also because it exhibits **nontrivial features** such as spontaneous symmetry breaking and second-order phase transitions.  Furthermore, it belongs to the same universality class as the $\phi^4$ lattice field theory.  However, following referees' suggestions, we are working on implementing new theories for testing our method.  We plan to upload a revision with new results before the discussion phase ends.

- **Mathematical Formulation and pseudocode**: Several referees complained about the clarity of the mathematical expressions, in particular about the sequential training procedure of RiGCS.  We improved the clarity by adding explanations, illustrations of the workflow, as well as  pseudocodes in Appendix F.6.

---

> ### Author Response · Authors · 2024-11-22
> **References**
>
> - [1] [Cranmer, Kyle, et al. "Advances in machine-learning-based sampling motivated by lattice quantum chromodynamics." Nature Reviews Physics 5.9 (2023): 526-535.](https://www.nature.com/articles/s42254-023-00616-w)
> - [2] [Nicoli, Kim A., et al. "Estimation of thermodynamic observables in lattice field theories with deep generative models." Physical review letters 126.3 (2021): 032001.](https://journals.aps.org/prl/abstract/10.1103/PhysRevLett.126.032001)
> - [3] [Kanwar, Gurtej, et al. "Equivariant flow-based sampling for lattice gauge theory." Physical Review Letters 125.12 (2020): 121601.](https://journals.aps.org/prl/pdf/10.1103/PhysRevLett.125.121601)
> - [4] [Wu, Dian, Lei Wang, and Pan Zhang. "Solving statistical mechanics using variational autoregressive networks." Physical review letters 122.8 (2019): 080602.](https://journals.aps.org/prl/abstract/10.1103/PhysRevLett.122.080602)
> - [5] [Li, Shuo-Hui, and Lei Wang. "Neural network renormalization group." Physical review letters 121.26 (2018): 260601.](https://link.aps.org/doi/10.1103/PhysRevLett.121.260601)

---

> ### Author Response · Authors · 2024-11-28
> **Revision upload and additional experiments on phi-4 theory**
>
> We sincerely thank all reviewers for the active discussion in the rebuttal phase.  We would like to inform the reviewers that we have uploaded a revision, where most of the reviewers’ suggestions have been incorporated, except some parts related to additional experiments.  Revised parts except minor corrections, e.g., typos, are highlighted in blue.  A major revision was done in Section 3.3, where the sequential training with model transfer is explained.  We added Fig. 2 that intuitively explains the procedure, including how the trained model parameters are used to initialize the parameters in the subsequent steps (levels).   Due to the space limit, we moved a large part of the mathematical descriptions to Appendix E.1, where the explanations on how the parameters are initialized and on how SIC justifies the sequential training process are further elaborated.  Pseudocode for training and sampling is also added there.
>
> Some experimental parts have not been fully updated yet as they will depend on the outcome of the suggested additional experiments, for which we could only conduct preliminary experiments so far.  Specifically, we implemented a continuous version of RiGCS (as well as VAN) with continuous autoregressive neural network (ARNN) using a mixture of Gaussians [1], and applied it to the two-dimensional $\phi^4$ lattice field theory up to $N=32$, see the [**ANONYMIZED LINK**]( https://www.dropbox.com/scl/fi/rkw36rtgganudhden5lw7/phi4ESS.pdf?rlkey=2qc1b3qo3iyzzvzeixjmw79tj&st=cyd3y3g9&dl=0).
> Similarly to the Ising model case, RiGCS significantly outperforms the plain VAN, which could be trained only up to $N=16$ within reasonable computation time.  Note that the ESS of both models are worse in the continuous $\phi^4$ theory compared to the discrete Ising model (Fig. 4 right),  not only because the theory is harder, but also because the continuous ARNN might not be the optimal choice as the basic building block.  We conjecture that RiGCS based on other models, e.g., affine-coupling normalizing flows and continuous normalizing flows, could significantly improve the ESS. We would like to leave this exploration to our future work.
>
>
> ## Refs.
>
> - [1] [Uria, Benigno, et al. "Neural autoregressive distribution estimation." Journal of Machine Learning Research 17.205 (2016): 1-37.](https://arxiv.org/abs/1605.02226v3)

---

### Meta-Review · Area_Chair_aWGh · 2024-12-20

**Metareview:**

The paper considers the problem of sampling from discrete distributions (such as the Ising model or the Potts model) near the criticality regime where the scale invariance is present. This regime is notoriously difficult for Markov Chain Monte Carlo algorithms and, at the same time, is of great interest in natural sciences. To overcome this challenge, the authors propose to train a cascade of generative models with gradually increasing resolution where the samples from the coarser scales are used as a warm start for the finer scales. Finally, the authors study the proposed approach empirically.

Despite the fact that the proposed algorithm does not outperform cluster methods, the reviewers consider the contribution of the paper to be significant and relevant to the ICLR community.

**Additional Comments On Reviewer Discussion:**

Reviewers 1WHN, pv8Q, and cr5x found the initial empirical study limited. In response, the authors conducted an additional empirical study during the rebuttal, convincing the reviewers of the utility of the proposed algorithm. Reviewer cr5x highlighted the efforts of the authors to make the problem accessible for non-physicists.

---

### Decision · Program_Chairs · 2025-01-22

Accept (Poster)